# Astronomical tuning of the Aptian stage and its implications for age recalibrations and paleoclimatic events

C. G. Leandro [1✉], J. F. Savian [1,2], M. V. L. Kochhann [3], D. R. Franco [4], R. Coccioni [5], F. Frontalini [6], S. Gardin[7], L. Jovane [8], M. Figueiredo[9], L. R. Tedeschi[9], L. Janikian[10], R. P. Almeida[11] & R. I. F. Trindade [12]

The Aptian was characterized by dramatic tectonic, oceanographic, climatic and biotic changes and its record is punctuated by Oceanic Anoxic Events (OAEs). The timing and duration of these events are still contentious, particularly the age of the Barremian-Aptian boundary. This study presents a cyclostratigraphic evaluation of a high-resolution multiproxy dataset ($\delta^{13}$C, $\delta^{18}$O, MS and ARM) from the Poggio le Guaine core. The identification of Milankovitch-band imprints allowed us to construct a 405-kyr astronomically-tuned age model that provides new constraints for the Aptian climato-chronostratigraphic framework. Based on the astronomical tuning, we propose: (i) a timespan of ~7.2 Myr for the Aptian; (ii) a timespan of ~420 kyr for the magnetic polarity Chron M0r and an age of ~120.2 Ma for the Barremian—Aptian boundary; and (iii) new age constraints on the onset and duration of Aptian OAEs and the 'cold snap'. The new framework significantly impacts the Early Cretaceous geological timescale.

[1] Programa de Pós-Graduação em Geociências, Universidade Federal do Rio Grande do Sul, Avenida Bento Gonçalves, 9500, 91501-970 Porto Alegre, RS, Brazil. [2] Departamento de Geologia, Instituto de Geociências, Universidade Federal do Rio Grande do Sul, Avenida Bento Gonçalves, 9500, 91501-970 Porto Alegre, RS, Brazil. [3] Instituto de Geociências, Universidade de São Paulo, Rua do Lago 562, 05508-080 São Paulo, SP, Brazil. [4] Coordenação de Geofísica, Observatório Nacional, R. General José Cristino, 77, 20921-400 Rio de Janeiro, RJ, Brazil. [5] Università degli Studi di Urbino "Carlo Bo", 61029 Urbino, Italy. [6] Dipartimento di Scienze Pure e Applicate, Università degli Studi di Urbino "Carlo Bo", Campus Scientifico, Località Crocicchia, 61029 Urbino, Italy. [7] CR2P— Centre de Recherche en Paléontologie—Paris, UMR 7207, Sorbonne Université–MNHN–CNRS, 4, Place Jussieu, 75005 Paris, France. [8] Instituto Oceanográfico, Universidade de São Paulo, Praça do Oceanográfico 191, 05508-120 São Paulo, SP, Brazil. [9] Centro de Pesquisas e Desenvolvimento Leopoldo Américo Miguez de Mello, Petrobras Petróleo Brasileiro S.A, Avenida Horácio Macedo 950, 21941-915 Rio de Janeiro, Brazil. [10] Departamento de Ciências do Mar, Instituto do Mar, Universidade Federal de São Paulo, Rua Carvalho de Mendonça, 144, 11070-102 Santos-SP, Brazil. [11] Departamento de Geologia Sedimentar e Ambiental, Instituto de Geociências, Universidade de São Paulo, Rua do Lago 562, 05508-090 São Paulo, Brazil. [12] Departamento de Geofísica, Instituto de Astronomia, Geofísica e Ciências Atmosféricas, Universidade de São Paulo, Rua do Matão 1226, 05508-090 São Paulo, SP, Brazil. ✉email: carolina.leandro@ufrgs.br

The Aptian is the third-longest stage in the Cretaceous (121.4–113.2 Ma)[1]. It is a critical time interval for widespread changes in plate dynamics, the carbon cycle, and the ocean–climate system (e.g., ref. [2]). It was characterized by globally significant episodes of marine carbon burial, known as Oceanic Anoxic Events (OAEs)[3], as well as significant evaporite deposition in the South Atlantic and changes in sulfur, calcium, strontium, and osmium in the oceans[4,5]. It has been suggested that a dramatic increase in the global mean temperature occurred during the Aptian, in which OAE 1a (~120 Ma) represents one of the most prominent events[6]. However, a period of temperature decrease has also been reported in the Late Aptian—the so-called "cold snap"[7,8]. Changes in the oceanic biota occurred throughout the Aptian, including significant turnovers in the late Aptian and across the Aptian−Albian boundary[9–11]. The Aptian stage is also marked by the onset of rapid seafloor spreading in the Atlantic Ocean caused by increasing oceanic crust production and plate margin volcanism[2]. It is also of great interest from a geodynamic or paleomagnetic perspective due to its association with the onset of the Cretaceous Normal Superchron (CNS; ~83–121 Ma)[12]. During the CNS, Large Igneous Province (LIP) activity peaked with the development of the Ontong Java Plateau (OJP), which is considered to be the driving force for widespread oceanographic changes at the onset of magnetic polarity Chron M0r[2,6,13,14]. These changes may have been responsible for the deposition of a thick, laminated, organic-rich black shale interpreted as the sedimentary expression of OAE 1a, known as the Selli Level, in Umbria-Marche Basin, central Italy[6,15,16].

Organic-rich horizons correlated to the Selli Level have been recognized in marine deposits worldwide, indicating that OAE 1a was a global event (e.g., refs. [1,3,6]). Re-Os radiometric ages obtained from the black shales of the Cismon core placed this event at 120 ± 3.4 Ma[16]. Several radiometric ages have been obtained for the OJP and associated basalt flows (e.g., refs. [17,18]). These cluster around the age of the Selli Level (~126–119 Ma), which, together with changes in the metal content, osmium and lead isotopes of ocean sediments[6,18,19], reinforces a possible causal link with the OJP. In addition to the Selli, others distinctive organic-rich black shale and calcareous mudstone marker beds occur within the Aptian interval, recognized mainly in the Tethyan realm. From bottom to top, they are: (1) the Wezel Level horizon[20] (2) the Fallot Level (~117.8 Ma[21,22]), (3) the 113/Jacob Level (~113–115 Ma[21,23]), and (4) the Kilian Level (~112–113 Ma), the last one marking the Aptian−Albian boundary[7,11,21,24]. Although the definition of black-shale levels corresponding to the OAE 1b differs, depending on the study, the Jacob and Kilian Levels are commonly accepted as records of organic-rich expressions of the first two sub-events of OAE 1b[25].

The timings and durations of the Aptian stage and its main events, such as the OAEs, are still debated[26]. The top of the Aptian is well defined at 113.1 ± 0.3 Ma on the basis of the U-Pb dating on zircons from an ash layer at Vöhrum, northern Germany[1,27]. The base of the Aptian is placed at 126 ± 0.3 Ma[28] in the Geologic Time Scale 2012 (GTS 2012). It coincides with the base of Chron M0r, and is supported by planktonic foraminiferal, calcareous nannofossil, and cyclostratigraphic data[21]. In contrast, $^{40}Ar$–$^{39}Ar$ ages (121.2 ± 0.5 Ma) from M0r lavas in China[29], and U-Pb ages (~121–122 Ma) in Barremian ash layers from Svalbard, imply that the Barremian−Aptian boundary must be younger than 123 Ma[30]. In addition, a new magnetostratigraphy of core DH1 from Svalbard (Norway) and the U-Pb dating of a bentonite bed at 123.1 ± 0.3 Ma in the uppermost part of the magnetozone M1r, also suggest a younger Barremian–Aptian boundary. The M1r magnetozone is ~1.9 Myr older than the onset of Chron M0r[31]. This interpolation-based constraint places the beginning

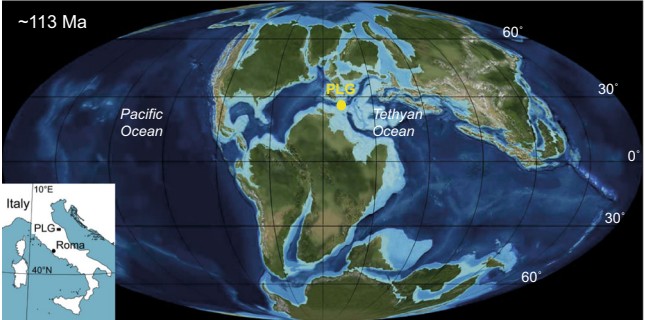

**Fig. 1 Paleogeographic reconstruction at 113 Ma.** Location of the Poggio le Guaine (PLG) core in the Tethyan realm at 113 Ma (www.jan.ucc.nau.edu).

of Chron M0r at 121.2 ± 0.4 Ma, therefore ~5 Myr younger than its age in the GTS 2012[1,31].

Floating timescales derived from cyclostratigraphic studies have been used to reach divergent conclusions. An age of ~121 Ma is suggested for the Barremian−Aptian boundary for the Italian Cismon APTICORE, an incomplete section of the Aptian that shows an unconformity in the late Aptian[24]. Conversely, a greyscale cyclostratigraphic study for the Piobbico core calibrates the Aptian at 13.42 Myr, providing age of 125.45 Ma for the Barremian−Aptian boundary[21]. Therefore, floating timescale from other complete Aptian records might improve this ambiguous scenario.

Here, we present a cyclostratigraphic analysis of the Poggio le Guaine (PLG) core[32] (Fig. 1), comprising a complete Aptian record from the Umbria-Marche Basin (UMB), Italy. The sedimentary succession covers the M0r and C34n (CNS) magnetozones throughout the late Barremian to early Albian[33]. Our main objectives are: (i) to propose a cyclostratigraphic framework for the PLG section using high-resolution magnetic susceptibility (MS), anhysteretic remanent magnetization (ARM), and oxygen (δ18O), and carbon (δ13C) stable isotopes data to provide better constraints for the Aptian climato-chronostratigraphic framework; and (ii) to discuss the impact of the proposed framework on the main events of the Aptian and the Cretaceous timescale.

## Results

**Magnetic and stable isotope variations**. MS and ARM vary between $1.1 \times 10^{-9}$ m³/kg and $5.66 \times 10^{-7}$ m³/kg and from $4.51 \times 10^{-11}$ Am²/kg to $1.16 \times 10^{5}$ Am²/kg, respectively. The highest MS and ARM values occur in clay-, marl-, and limestone-rich lithofacies of the Marne a Fucoidi Formation. In contrast, the lowest values occur in the white lithofacies of the Maiolica Formation (comprising mainly limestone and marly limestone/calcareous marl). Significant low- and high-wavelength quasiperiodic contents can be observed for both the MS and ARM datasets (Fig. 2a, b). Rock magnetic measurements reveal magnetite as the main magnetic carrier[33].

The δ13C values vary significantly in the Aptian (1.47–4.82‰; Fig. 2c). The 95.99 m (base)–91.19 m interval displays δ13C values between 1.97 and 3.67‰. The 95.12 m (Barremian–Aptian boundary) to 91.19 m corresponding to the C1, C2 and Ap1, Ap2 segments[7,33]. The Selli Level (91.19–89.24 m) occurs at the base of the PLG core (C3 to C6 or Ap3 to Ap6)[32,33]. The δ13C-negative excursion in C3 (1.47‰) followed by a positive excursion (up to 4.44‰) unambiguously indicates OAE 1a. However, the PLG core is virtually carbonate-free in this interval, and the sparse δ13C data do not allow us to define the two positive excursions (C4 to C6 and Ap4 to Ap6 segments) defined globally[7,33]. The C7/Ap7 segment[7] corresponds to higher δ13C values (3.65–4.82‰) at 89.24–86.12 m. The Wezel Level[20]

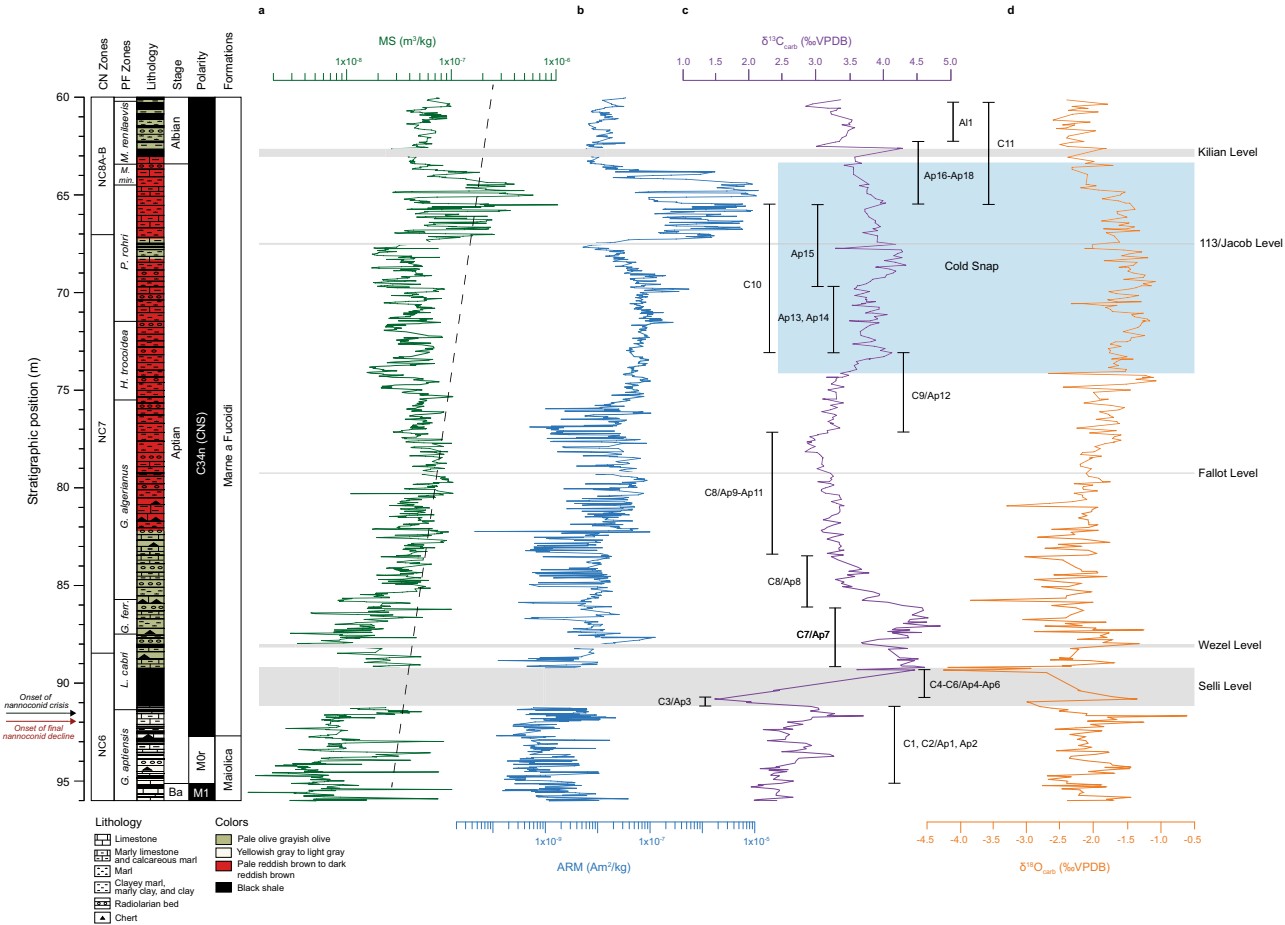

**Fig. 2 Integrated stratigraphy of the studied interval at PLG core.** Stratigraphic framework of the PLG core with stratigraphic depths. Depths for the upper boundaries of the planktonic foraminiferal and calcareous nannofossil zones, and nannoconid decline and crisis biohorizons identified in the PLG core[11,32] and this work. Changes in (**a**) magnetic susceptibility (MS) (dark green) with detrending linear (black-doted line); **b** anhysteretic remanent magnetization (ARM) (blue); **c** $\delta^{13}C$ (purple). Codes for C/Ap-isotope segments[7,33]. **d** $\delta^{18}O$ (orange). The gray bands highlight the Selli, Wezel, Fallot, 113/Jacob and Kilian Levels. The blue band in the stable isotope data represents the cold snap interval. Ba. Barremian, *M. Microhedbergella, min. miniglobularis, P. Paraticinella, H. Hedbergella, G. Globigerinelloides, ferr. ferreolensis, L. Leupoldina.*

(88.20–88.00 m) is marked by a slight $\delta^{13}C$-negative excursion from 4.33‰ to 3.65‰ within the C7/Ap7 segment. The intervals from 89.24–88.20 m and 88.00–86.10 m in the C7/Ap7 segment are a pale greyish-olive color; the feature persists up to 82.10 m. From 86.12 to 78.18 m, $\delta^{13}C$ values decrease from >4‰ to 2.85‰, characterizing the C8/Ap8 (86.12–83.51 m) and C8/Ap9–Ap11 (83.36–78.18 m) segments[7]. Pale to dark reddish-brown limestones and clay marls are the predominant lithology in the interval 82.10–63.02 m. The Fallot Level occurs in the 79.37–79.31 interval within C8/Ap9–Ap11 segments[7]. The C9/Ap12 segment in the interval 78.18–73.00 m shows a characteristic increase in $\delta^{13}C$ values from 2.81 to 4.10‰ and defines the onset of a broad positive carbon isotope excursion in the late Aptian, which continues in the C10/Ap13-Ap15 segment. $\delta^{13}C$ values are mostly over 3.5‰ in the interval 73.00–67.70 m. A slightly increasing $\delta^{13}C$ trend culminates in a prominent peak shift (3.28‰) at 67.70 m. The 113/Jacob Level (67.44–67.52 m) is marked by a positive peak at 4.16‰ subsequently decreasing to 3.26‰ within the C10/Ap15 segment[7]. The C11/Ap16–Ap18 segment (65.42–62.26 m) shows $\delta^{13}C$ values decreasing from ~4 to 3.39 ‰, with rapid increasing $\delta^{13}C$ values and a positive shift to 4.26‰ at the top of the black-shale Kilian Level (62.64 m). The stratigraphically higher (63.02–60.14 m) section comprises pale olive to greyish-olive marls, clayey marl, marly clay, and clay. Here, $\delta^{13}C$ values vary slightly from 2.82 to 3.55‰

and correspond to the C11/Al1 segments[7]. The $\delta^{13}C$ profile shows an ambiguous correlation to others Aptian records worldwide such as Santa Rosa Canyon in Mexico[34] and DSDP Site 463 in the Pacific Realm[35] (Supplementary Fig. 1). Although we cannot totally exclude diagenetic overprint on each $\delta^{13}C$ measurement, the correlation of $\delta^{13}C$ profile with sections worldwide and the presence of all C/Ap segments together with well–none magnetostratigraphy and biostratigraphy suggest an unambiguously reliability on the PLG $\delta^{13}C$ profile as close to palaeoenvironmental proxy rather than a general section with diagenetic overprint. It also testifies the completeness of PLG isotopic and lithological records. Moreover, the lack of covariance between $\delta^{13}C$ and $\delta^{18}O$ also suggests minimal diagenetic overprint on $\delta^{13}C$, as $\delta^{18}O$ are usually affected by diagenetic overprint in marine environment[35].

The $\delta^{18}O$ values range from −4.22 to −0.60‰ (Fig. 2d). At the core's base (light gray to yellowish-gray marl, clayey marl, marly clay, and clay lithologies), $\delta^{18}O$ values range from −2.75 to −1.25‰, with a prominent peak (−0.63‰) at 91.65 m. The onset of the Selli Level is marked by a slight decrease in $\delta^{18}O$ (−2.98‰), corresponding to the negative excursion in $\delta^{13}C$ (segment C3/Ap3). In the central part of the Selli Level (segments Ap5/C5–Ap4/C4), $\delta^{18}O$ values fluctuate between −2.69 and −1.34‰. The Ap6/C6 segment shows decreasing $\delta^{18}O$, fluctuating between −4.15 and −2.92‰ with a major peak (−4.22‰) at

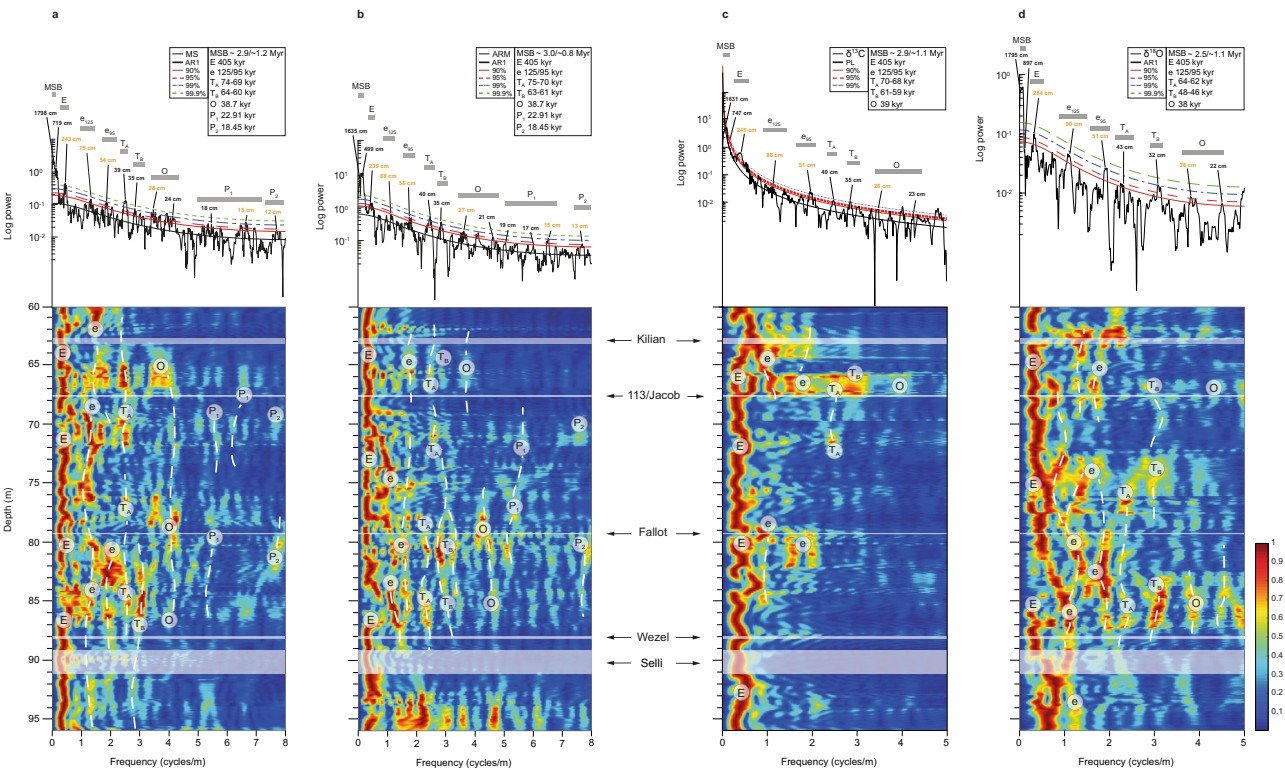

**Fig. 3 Spectral analysis of the cyclostratigraphic series. a** Magnetic susceptibility (MS); **b** anhysteretic remanent magnetization (ARM); **c** δ13C; **d** δ18O. Top: 2π multitaper power spectra, with the AR(1) red noise spectral model and 90%, 95%, 99%, and 99,9% confidence levels (c.l.) for null hypothesis testing. Wavelengths of spectral peaks are labeled in cm. Bottom: evolutionary fast Fourier transform (eFFT) spectrograms with a 4 m sliding window, with each calculated eFFT spectrum normalized to 1. MSB million-year scale band, E long eccentricity, e125 and e95 125-kyr and 95-kyr short eccentricity, $T_A$ and $T_B$ are referred to periodicities of ~60–70 kyr, O obliquity, $P_1$ and $P_2$ precession.

89.34 m. In the stratigraphic interval 88.00–79.37 m, δ18O values oscillate between −3.10 and −1.30‰, with two peaks at 85.75 m and 80.90 m (−3.88‰ and −3.30‰, respectively). Above 79.30 m, the δ18O values increase slightly, reaching a maximum of −1.10‰ at 74.32 m. The trend is interrupted by a negative shift to −2.70‰ at 74.10 m, followed by higher δ18O values (−2.35 to 1.10‰) in the interval 74.10–63.35 m that correspond to lower surface water temperatures during the late Aptian cold snap[7,8]. Above 63.35 m, δ18O values decrease from −1.78 to −2.54‰ and suggest a warming trend after the cold snap. The δ18O values from bulk rock are usually associated to diagenetic overprint. In our PLG section, diagenetic overprint probably has taken place on all sections in order to turn δ18O profile noisy and δ18O values relatively depleted than a pristine fossil record. Nevertheless, the relatively higher δ18O values in the interval in the interval 74.10–63.35 m matches the same stratigraphic correlation observed in Piobicco core (Italy) and DSDP Site 463 (Pacific Realm)[7] based on bio- and carbon isotope stratigraphy, where nannofossil proxies together with relatively higher δ18O values suggest relatively colder temperatures during C9 to C11 segment in the late Aptian. Moreover, it also shows good agreement to DSDP Site 545[8] based on the same bio- and carbon isotope stratigraphy, where cold snap has been identified based on TEX86 proxies, which are specific organic compounds. Therefore, the relatively higher δ18O values in our section have been interpreted as a "cold snap" record.

**Aptian astronomically tuned chronology.** We identified spectral patterns suggesting astronomical forcing for all four datasets (MS, ARM, δ13C, δ18O; Fig. 3a–d). Spectral analyses (Fig. 3, top) revealed statistically significant spectral peaks above the 95%

confidence level. Comparing these spectral peak frequencies with evolutionary fast Fourier transform (eFFT) results (Fig. 3, bottom), we observed a pronounced low-wavelength stability pattern throughout the section. The wavelength ratios verified for specific spectral peaks at the MS (243:75:54:28:15:12 = 20.25:6.25:4.50: 2.33:1.25:1), ARM (239:88:56:27:15:13 = 18.38:6.77:4.31:2.08:1.15: 1), δ13C (245:88:51:28 = 18.4:6.6:3.8:2.1) and δ18O (284:90:51: 26 = 23:7.3:4.1:2.1) spectra resemble the predicted Milankovitch spectral peak ratios for Albian–Aptian times[36] (405:125:95: 38.7:22.91:18.45 = 22.0:6.8:5.1:2.1:1.2:1). Due to their lower sampling resolution (~10 cm), the stable isotope records do not show the precession signal, which is below the Nyquist frequency. We interpret the spectral peak bands for the magnetic data (MS and ARM) of 12–13 cm and 14–19 cm as the precession parameters $P_2$ (~18.5 kyr) and $P_1$ (~22.9 kyr). We interpret obliquity for the spectral bands of 23–29 cm (MS), 21–29 cm (ARM), 22–29 cm (δ13C), and 22–28 cm (δ18O), corresponding to ~38.7 kyr (MS and ARM), ~39 kyr (δ13C) and ~38 kyr (δ18O). It is worth mentioning that recognizing this cyclicity pattern on δ18O could be interpreted as either paleoenvironmental or diagenetic cycles. Although we cannot rule out the diagenetic overprint, it would be reasonable to observe that cycles from matches other δ18O paleoenvironment proxies used in this study. Thus, it is more likely that δ18O cycles represent some paleoenvironmental changes, whether it has suffered diagenetic overprint or not.

The 125-kyr and 95-kyr short eccentricity signals (marked "e" in Fig. 3; see also the yellow bundles in Fig. 4c for MS data) could be the spectral bands of 45–58 cm and 68–104 cm (MS), 52–64 cm and 80–107 cm (ARM), 47–58 cm and 72–104 cm (δ13C), and 50–58 cm and 67–105 cm (δ18O). The short eccentricity cycle is also evident in the MS record through the ~4:1 bundling of short eccentricity to precession (yellow bundles

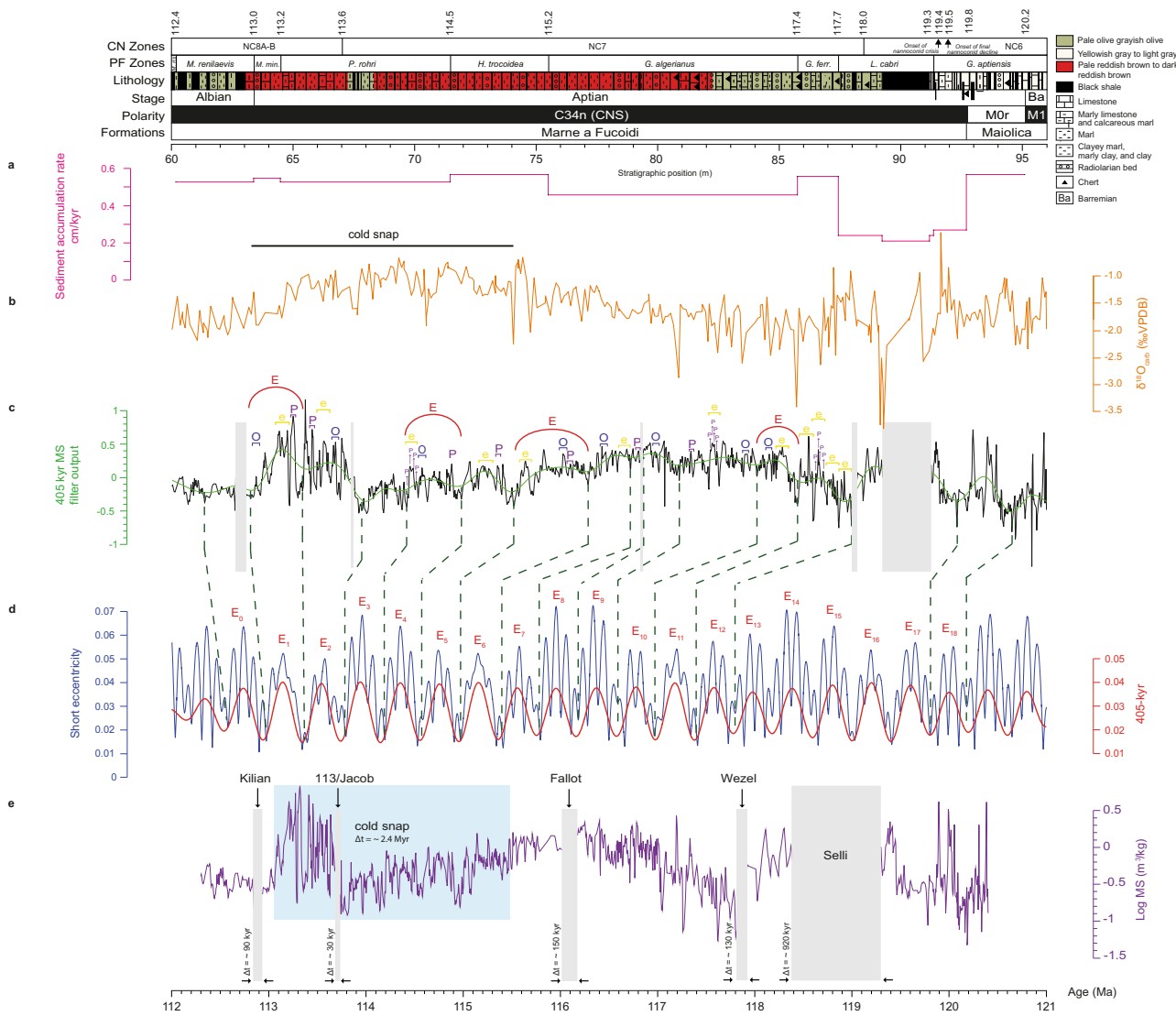

**Fig. 4 Astronomical calibration of the PLG. a** Sediment accumulation rate (SAR) curve based on 405-kyr tuning (pink); **b** $\delta^{18}O$ (orange); **c** logarithmic-scale magnetic susceptibility (MS) data with linear interpolation and detrending linear (black line) and 405 kyr filter output (green line); **d** La2004 orbital solutions[39] for the eccentricity cycles of ~100 kyr (blue line) and ~405 kyr (red line); **e** data calibrated by long eccentricity (purple). The gray bands highlight the Selli, Wezel, Fallot, 113/Jacob and Kilian Levels and the blue band represents the cold snap interval. Formations, lithology and planktonic foraminiferal zones[11,32,45] and this work, calcareous nannofossil zones (this work), polarity[33] and stage (this work) for the PLG core. *M. Microhedbergella, ris. rischi, min. miniglobularis, P. Paraticinella, H. Hedbergella, G. Globigerinelloides ferr. ferreolensis, L. Leupoldina.* E long eccentricity (red bundles), e short eccentricity (yellow bundles), O obliquity (blue bundles), P precession (purple bundles).

in Fig. 4c). The spectral content confined to the 191.3–321.2 cm (MS), 181.7–276.7 cm (ARM), 188.7–338.4 cm ($\delta^{13}C$), and 190.8–359.0 cm ($\delta^{18}O$) spectral bands could be related to the ~405-kyr eccentricity cycle (herein referred as "E"). Cyclostratigraphic analysis for the MS dataset (Fig. 3a, top and bottom) indicates a ~243 cm wavelength above the 99.9% confidence level (c.l.), which is applied for the astronomical tuning also indicated as red bundles in Fig. 4c.

Periodicities of ~60–70 kyr occur in both spectra (herein referred to as "$T_A$" and "$T_B$" spectral ranges). These are particularly evident from the eFFT for the MS dataset (Fig. 3c) in the upper first (~82–87 m) and last (~65.5–67.0 m) thirds of the PLG core interval. Despite the detrending, low-frequency spectral peak ranges in the Myr-scale wavelength (at ~1798–640 cm (MS), ~1635–449 cm (ARM), ~1631–598 cm ($\delta^{13}C$), and ~1795–1380 cm ($\delta^{18}O$)) were also verified above a

99.9% confidence level (MS data are shown as black bundles in Fig. 4c). These features could be linked to non-periodic long-term climatic variations or long-term orbital modulation cycles in sedimentary profiles worldwide. Therefore, they could indicate a significant control on the global climate (e.g., refs. [37,38]).

Using a low-pass filter with a cutoff frequency of 0.4 cycles/m, we isolated the interpreted 405-kyr long-eccentricity component from the MS dataset. Since there are no available radiometric ages for the Marne a Fucoidi Formation, we assigned the stratigraphic position of the Aptian–Albian boundary to ~113.0 Ma[1,27]. Therefore, we built the ~405-kyr tuned age model for the PLG core based on the long-eccentricity low-pass filter output from MS data (Fig. 4c, in green) and the g2-g5 target curve from the La2004 astronomical solution[39] for Aptian–Albian times (Fig. 4d, in red). This procedure allowed us to build a floating astronomical timescale (ATS) based on 18 long-eccentricity

**Table 1 Estimated timespan of Aptian events.**

| Work | Timespan (kyr) | | | | |
|------|------|------|------|------|------|
| | Selli | Wezel | Fallot | 113/Jacob | Kilian |
| Malinverno et al.[24] | 1110 ± 0.11 | | | | |
| Huang et al.[21] | 1400 | | ~360 | ~40 | ~120 |
| This work | ~920 | ~130 | ~150 | ~30 | ~90 |

Comparison of a timespan for the Aptian-earliest Albian black-shale levels[21,24] and this work.

**Table 2 Estimated timespan of bioevents.**

| Biostratigraphic zones | Depth (m) | Age (Ma)* | Timespan (Myr) |
|------|------|------|------|
| M. renilaevis | 60.20 | 112.4 | 0.6 |
| M. miniglobularis | 63.40 | 113.0 | 0.2 |
| P. rohri | 64.50 | 113.2 | 1.3 |
| NC7 (R. angustus) | 67.00 | 113.6 | 4.4 |
| H. trocoidea | 71.47 | 114.5 | 0.7 |
| G. algerianus | 75.51 | 115.2 | 2.2 |
| G. ferreolensis | 85.76 | 117.4 | 0.3 |
| L. cabri | 87.44 | 117.7 | 0.6 |
| NC6 (C. litterarius) | 88.44 | 118.0 | – |
| G. aptiensis p.p. | 91.35 | 119.3 | 0.9 |
| Onset of nannoconid crisis | 91.54 | 119.4 | – |
| Onset of final nannoconid decline | 91.94 | 119.5 | – |

Depths for the upper boundaries of the planktonic foraminiferal and calcareous nannofossil zones, and nannoconid decline and crisis biohorizons identified in the PLG core[11,32,45] and this work, and their ages and timespan estimated in this work.
*Top of the biostratigraphic zone.

cycles and an age model for the Aptian interval of the PLG core comprising ~7.2 Myr from the lowest occurrence (LO) of *Microhedbergella renilaevis* and the base of Chron M0r (Fig. 4d). From the MS data age model, we can infer durations of ~920 kyr, ~130 kyr, ~150 kyr, ~2.4 Myr, ~30 kyr, and ~90 kyr for the Selli, Wezel, Fallot, cold snap, 113/Jacob, and Kilian Levels, respectively (Table 1). Our study also provides new planktonic foraminifera and calcareous nannofossil data for the PLG core (Fig. 4 and Table 2).

Evaluation of sediment accumulation rate (SAR) throughout the PLG by means of correlation coefficient (COCO) and evolutionary correlation coefficient (eCOCO)[40] (Supplementary Figs. 2–4) confirm the previous low-resolution SAR derived from biostratigraphic datums as well as the higher resolution SAR derived from our astronomical tuning (~0.43 cm/kyr) (Fig. 4a).

## Discussion

The 405-kyr tuned age model for the PLG core is based on the g2-g5 target curve from the La2004 solution for the Aptian–Albian. It suggests that the 36 m section of the PLG core represents ~8.1 Myr of sedimentation (120.4–112.3 Ma in the interval 96–60 m). Multiproxy spectral analysis showed similar power spectra (Supplementary Fig. 5). This remarkable pattern from different proxies provides a reliable cyclostratigraphic evaluation for the Aptian. All proxies exhibit strong signals of the 405-kyr eccentricity, the most important signal for tuning an ATS due to its stability far back in time[41].

So far, the only estimate of the mean sedimentation rate inferred for the entire Aptian (based on the PLG core) in literature is ~0.24 cm/kyr[32]. In contrast, our sedimentation rate curve (Fig. 4a) indicates a mean SAR of 0.43 cm/kyr for the most of Aptian. This result is consistent with the COCO/eCOCO results (two peaks of 0.52 cm/kyr and 0.58 cm/kyr associated to a null hypothesis significance level lower than 0.001—Supplementary Figs. 2–4).

Furthermore, our constructed floating ATS provides a continuous "absolute time framework" for the Aptian interval (from 63.4 to 95.1 m), indicating a ~7.2 Myr duration for the PLG core based on 18 long-eccentricity cycles (E$_1$–E$_{18}$, Fig. 4d).

Previous cyclostratigraphic studies of the Marne a Fucoidi Formation constrained the duration of the Aptian to 6.4 ± 0.2 Myr in the section from central Italy[42]. The same lithostratigraphic unit dominates the PLG core (92.70–60.00 m), consisting of a pelagic carbonate succession with a low sedimentation rate. A cyclostratigraphic analysis of shallow-water carbonate deposits in South Italy indicated an Aptian stage duration of about 7.2 Myr[43]. An alternative calibration of 6.8 ± 0.4 Myr was obtained based on astrochronology and constrained by direct absolute dating of ten glauconitic horizons in the Late Hauterivian–Early Albian section from the Vocontian basin (southeast France)[44]. Therefore, at least three sections in Italy and France give similar timespans (6.4 ± 0.2 Myr to 7.2 Myr)

to this study (~7.2 Myr). However, a significantly longer 12.9 Myr Aptian duration has been widely used in GTS geological time-scales from 2004 to 2016 and international stratigraphic charts. This cyclostratigraphic study was based on supposedly 405-kyr cycles derived from a greyscale scan, an indirect proxy of the Aptian from the Piobbico core in central Italy[21] (Fig. 5a), and estimated an age of 125.45 Ma for the base of Chron M0r, modified to 126.3 Ma given a shift in the age of the base in GTS 2012[28]. We note that this astronomical calibration is tied to the Albian–Cenomanian boundary at 99.6 Ma and no further radiometric age ties.

The duration of the lower Aptian can also be estimated by addressing the inverse problem using the Formation microimager as a proxy for changes in the sedimentation rate for the Selli Level in the Cismon APTICORE. This method gives a duration of 1.11 ± 0.11 Myr for OAE 1a[24] (Fig. 5b). With the sampled age models tied to the base of Chron M0r at 121 Ma[23], the most likely interval for the Selli Level is 120.21–119.11 Ma, similar to the interval in this study (119.30–118.38 Ma).

Although this study does not present any new radiometric ages, we tied our section to the Aptian–Albian boundary at ~113.0 Ma, as defined by the GTS 2020[1,27]. The Barremian−Aptian boundary's age is among the most poorly constrained stratigraphic boundaries[18], despite several multiproxy efforts (e.g., refs. [14,18,21]). Some suggest that the boundary be placed near the top of the *Globigerinelloides aptiensis* Zone at the Gorgo a Cerbara section[45]. However, the top of the *G. aptiensis* planktonic foraminiferal Zone is ~3.75 m above the base of the M0r in the PLG core. Here, we assumed the Barremian–Aptian boundary was equivalent to the onset of M0r (e.g., refs. [18,21]). Therefore, we used the base of Chron M0r at 95.10 m in the PLG core to define the Barremian−Aptian boundary for the astronomical tuning. These tie points and our age model for the PLG core (~7.2 Myr) define an age of ~120.2 Ma for the base of M0r (Fig. 5c). This age is compatible with radiometric ages recently proposed for the M0r: (i) a $^{45}$Ar–$^{39}$Ar radiometric age of 121.2 ± 0.5 Ma for lava flows at the Mashenmiao-Zhuanchengzi (MZ) section, NE China;[29] (ii) a U-Pb zircon dating of Barremian tephra layers at 121–122 Ma;[30] (iii) the base of Chron M0r at 121.4 Ma from GTS 2020;[1] and (iv) an interpolated age of 121.2 ± 0.4 Ma for the M0r, from U-Pb radiometric ages for a bentonite bed in Norway[31].

Multiproxy orbital tuning for the PLG core gave an age of ~120.2 Ma for the Barremian–Aptian boundary, which is a better match with the timescale placing the base of the Aptian at

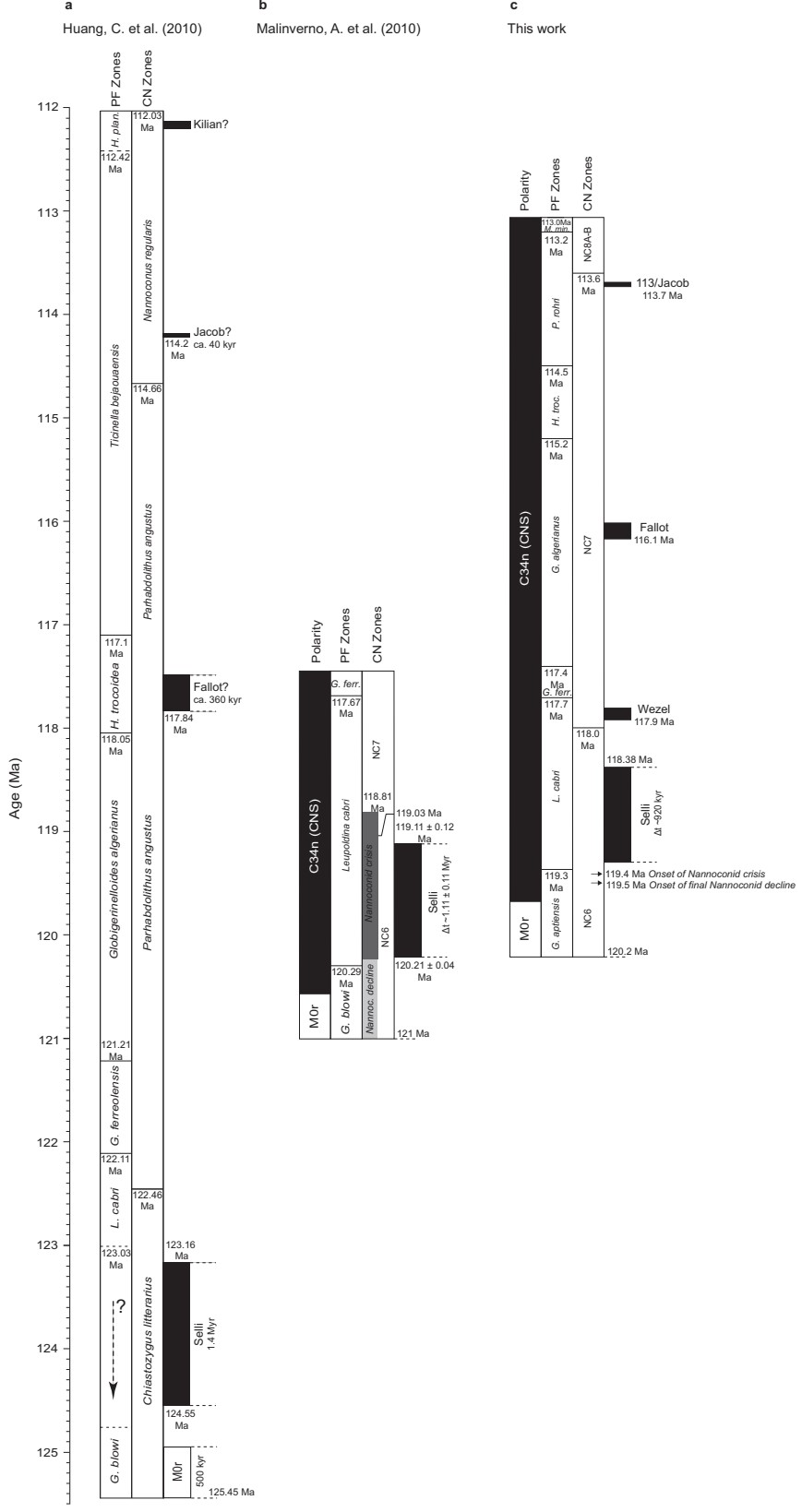

**Fig. 5 Comparison with Aptian timescales.** Stratigraphic correlation of **a** Huang et al.[21]; **b** Malinverno et al.[24]; **c** this work. The comparison tables indicate the magnetic polarity and planktonic foraminiferal and calcareous nannofossils zones. Stratigraphic extent of Selli Level equivalent sediments deposited during OAE 1a, Wezel Level, Fallot Level, 113/Jacob Level, and the Aptian–Albian boundary. The black arrows recorded decline and crisis of nannoconids. All stratigraphic scales are in age. *H. Hedbergella, plan. planispira, G. Globigerinelloides L. Leupoldina, ferr. ferreolensis, Nannoc. Nannoconid, M. Microhedbergella, min. miniglobularis, P. Paraticinella.*

**Table 3 Estimated ages of the black-shale levels.**

| Work | Black-shale levels (estimated age in Ma) | | | | |
|---|---|---|---|---|---|
| | Selli | Wezel | Fallot | 113/Jacob | Kilian |
| Malinverno et al.[24] | ~119.11 (U) ~120.21 (L) | | | | |
| Huang et al.[21] | 123.16 (U) 124.55 (L) | | 117.8 | 114.2 | 112.0 |
| Bottini et al.[17] | 120.4 ± 3.4 | | | | |
| Bottini et al.[7] | | | | | 113.0 |
| Sabatino et al.[23] | | | | 115.0 | 112.8 |
| This work | 118.8$_{119.30}^{118.38}$ | 117.9$_{117.93}^{117.80}$ | 116.1$_{116.17}^{116.02}$ | 113.7$_{113.72}^{113.69}$ | 112.9$_{112.93}^{112.84}$ |

Comparison of timespan for the Aptian-earliest Albian black-shale levels[21,24] and this work.
a$_L^U$: estimated mean age (a); U(L): estimated age for the upper (lower) boundary of the black shale.

121.54 Ma[14] in comparison to the age boundary (~125 Ma)[21,46] in the 2012 geologic time and international stratigraphic chart scales[27]. From a robust selection of U-Pb and $^{45}$Ar–$^{39}$Ar geochronology, it was proposed that the Barremian–Aptian boundary (considering the base of the Chron M0r) should be placed between 123.8 and 121.8 Ma[18]. In GTS 2020, the base of the Aptian is placed at 121.4 Ma[1]. Our results show a difference of ~1 Ma with the GTS 2020[1], compatible with recent U-Pb dates from a bentonite bed in Norway[31]. These results have direct implications for the time and duration of the M-sequence (Late Jurassic–Early Cretaceous) and, consequently, to the Geomagnetic polarity timescale (GPTS) for the Mesozoic Era, as the Barremian–Aptian boundary is a crucial tie-point for the M-sequence magnetic anomalies.

One limitation in understanding the relationship between OAEs and black-shale deposition during the Aptian–Albian is their uncertain timing and duration. The majority of paleoenvironmental and paleomagnetic studies focus on well-debated OAEs (e.g., OAE 1a and OAE 1b), while little is known about the interbedded black-shale levels between them.

Early Aptian sediments are marked by the black-shale horizons of OAE 1a (or Selli Level, UMB[15][15]). In the PLG core, organic-rich shales and increased total organic carbon (TOC) define the OAE 1a Level, which is also marked by a sharp negative (C3) and broad positive (C4–C6) excursions in $\delta^{13}$C[6,33]. The Selli Level is 195 cm thick (from 91.19 to 89.24 m) and occurs just above the Chron M0r (Figs. 2 and 4)[32,33]. It is identified in the lower-middle part of the *Leupoldina cabri* planktonic foraminiferal Zone. OAE 1a was defined in the PLG core based on Os and Hg isotopes[6,20].

Age constraints for the Selli Level are mainly based on magneto-biostratigraphy estimates;[47] its timespan is still far from certain. Our estimate of ~920 kyr, between the E$_{14}$ and E$_{16}$ long-eccentricity bundlings (Fig. 4d and Table 1), closely matches an estimate based on an inflection point in the $\delta^{13}$C curve at 1.11 ± 0.11 Myr[24]. It differs from previous estimates, e.g., 1.4 Myr[21] 1.0–1.3 Myr;[48] 1.157 Myr;[49] and 1.36 Myr[50]. Radiometric dating is also poorly constrained. Re-Os data from black shales in the Cismon core give an age of 120 ± 3.4 Ma[17], close to estimates for the base and top of the Selli Level (120.21 and 119.11 Ma, respectively)[24]. Here we estimated the ages for the Selli Level at 119.30 Ma (base) and 118.38 Ma (top) (Table 3), which closely match previous estimates and radiometric constraints. However, cyclostratigraphic analysis of the Piobbico core suggested ages of 124.55 Ma (base) and 123.16 Ma (top)[21].

Previous studies suggested ocean anoxia as a significant cause of the OAE 1a carbon-rich sediments, deposited ~0.5 Myr after the end of the M0r (e.g., ref. [51]). OJP basalt flows have been dated at 124–121 Ma by $^{39}$Ar–$^{40}$Ar ages[16], suggesting a causal link between OAE 1a and massive volcanism[19]. However, because mantle carbon has a $\delta^{13}$C composition of approximately −6‰

(e.g., ref. [52]), the negative $\delta^{13}$C excursion in the C3 segment (46.7 ± 13.7 ka)[24] requires too much volcanic $CO_2$ from the OJP. An alternative source of isotopically depleted carbon is the intrusion of magmatic sills ($\delta^{13}$C < −20‰) into organic-rich sediments. The High Arctic Large Igneous Province (HALIP) has recently been postulated as an alternative carbon source, producing $CO_2$ by metamorphism of organic matter (estimated at ~19,200 Gt of carbon). This source requires a lower emissions volume to cause the recorded negative $\delta^{13}$C excursion. The rapid release of aureole greenhouse gases (methane) from HALIP may have contributed to the negative $\delta^{13}$C excursion in OAE 1a from 120.2 ± 1.9 Ma to 124.7 ± 0.3 Ma[53]. However, the Hg-cycle perturbation by the HALIP in the onset of OAE 1a was smaller compared to the OJP[6]. A third possible source is the release of methane hydrates, a suggested trigger of the negative $\delta^{13}$C excursions from other OAEs, such as the Toarcian OAE. Based on our age estimate, the causal link between the onset of OAE 1a (C3 segment) and ages from OJP and HALIP suggests that these magmatic events helped to trigger the oceanic anoxia recorded in the Selli Level[6,53]. As the isotopic record of the C3 segment in PLG core do not show well-constrained boundaries to the lack of carbonate fraction, we were not able to add a further discussion about casual links between OAE 1a and possible sources of depleted $CO_2$.

The PLG core contains the Wezel Level, previously identified as the Noir Level[32]. It is probably traceable across the Tethyan region. It is located just above the top of the Selli Level from 88.2 to 88 m (Figs. 2 and 4). It has been identified in the upper part of the *L. cabri* planktonic foraminiferal Zone, near this common zonal marker, in the lowermost part of the NC7 calcareous nannofossil Zone. It occurs within the C7/Ap7 isotope zone with a relative negative carbon isotope excursion. Our results indicate a ~130 kyr duration (Table 1) and an average age of ~117.9 Ma (Table 3) for the Wezel Level.

We identify a sustained peak in stable isotopes in the PLG core between 67.52 and 67.44 m, interpreted as the so-called Fallot Level. This level is characterized by a series of black-shale layers formed under different forcing mechanisms, including enhanced burial of organic matter due to eutrophic conditions or low oxygenation at the seafloor[21]. According to some authors in the Tethyan realm (e.g., refs. [21,54]), four prominent black shales (FA 2', 2", 3 and 4) are recognizable.

The black shale FA3 is 8 cm thick in the PLG core and occurs in the *Globigerinelloides algerianus* planktonic foraminiferal and NC7 calcareous nannofossil Zones (Figs. 2 and 4). The timespan for this Fallot Level black shale is ~150 kyr (Table 1), and its estimated age is ~116.1 Ma (Table 3). Notably, a timespan of ~360 kyr at ~117.8 Ma was estimated for a ~1 m thick interval in the *Hedbergella trocoidea* planktonic foraminiferal and the NC7 calcareous nannofossil Zones in the Piobbico core (central Italy),

and it has been attributed to the Fallot Level[21]. However, black-shale layers are absent in the *Globigerinelloides ferreolensis*, *G. algerianus* and *H. trocoidea* planktonic foraminiferal Zones in the Piobicco core. Therefore, the presence of the Fallot Level in the Piobbico core is uncertain.

The main Paquier black-shale level in the Albian has been identified as the sole sedimentary expression of OAE 1b, but another three black-shale levels have been associated with OAE 1b: the 113/Jacob Level in the Aptian, the Kilian Level at the Aptian–Albian boundary, and the Lenhardt Level in the Albian[55]. The distinct black-shale horizons in the upper Aptian interval at PLG were interpreted as the 113/Jacob Level (e.g., refs. [11,32,55]).

However, the Jacob Level has only been documented in the Vocontian Basin at paleo-water depths between 500 and 1500 m[56]. This level is ~75 cm thick and lies in the *Ticinella bejaouensis* ( = *Paraticinella rohri*) planktonic foraminiferal and NC7B/NC7C calcareous nannofossil Zones in the Tarendol section (e.g., ref. [56]). In the Col de Pré-Guittard section, GSSP for the base of the Albian, the Jacob Level is 1.50 m thick and occurs in the *P. rohri* planktonic foraminiferal and in the NC7 calcareous nannofossil Zones[57,58].

The 113 level at PLG is an 8-cm-thick black shale in the upper part of the *P. rohri* planktonic foraminiferal and NC7 calcareous nannofossil Zones[11,32]. It is considered the sedimentary expression of the oldest subevent of OAE 1b and is thus equivalent to the Jacob Level. At DSDP Site 545, the lower of two negative carbon isotope excursions, labeled the HTE (high thermal event)[5], could be a North Atlantic equivalent of the Jacob subevent[11]. In the PLG core, the 113/Jacob Level occurs at a depth of 67.52 to 67.44 m (Figs. 2 and 4). The tuning presented herein suggests that the 113/Jacob Level is located at the eccentricity bundle $E_2$ (Fig. 4d). It has a duration of ~30 kyr (Table 1) and may represent one obliquity cycle. This result is compatible with the estimated ~40 kyr timespan for the Jacob Level in the Piobbico core[21]. Our estimated age for this level is ~113.7 Ma (Table 3), similar to previous estimates of ~114.2 Ma[21] and ~115 Ma[23].

The Kilian Level is recognized in the Vocontian Basin as a poorly fossiliferous, dark gray shale whose base is conspicuously bioturbated[59]. At Col de Pré-Guittard GSSP, this level is ~80-cm thick and occurs in the NC8A-B calcareous nannofossil Zone, with the Aptian–Albian boundary placed close to its middle part at the LO of the planktonic foraminifera *M. renilaevis*[57,58]. At the southern margin of the central–western Tethyan Ocean (UMB), the Kilian Level corresponds to a prominent black shale with very fine laminations and without bioturbation, which is 33 cm thick in the Piobbico core[60] and 38 cm thick in the PLG section and core[11,32]. In the PLG core, the Kilian Level occurs between 63.02 and 62.64 m and is represented by a poorly laminated black shale[32] (Figs. 2 and 4). In agreement with the findings at the PLG section[11], this level falls in the NC8A-B calcareous nannofossil Zone and the lowermost part of the *M. renilaevis* planktonic foraminiferal Zone. It is likely to be the earliest Albian in age (Figs. 2 and 4). According to our results, the Kilian Level, which occurs in the isotopic zone C11 or Ap16–Ap18[7], is located at the eccentricity bundle $E_0$ (Fig. 4d) and has a duration of ~90 kyr (Table 1). Based on our floating ATS, the age for this level is ~112.9 Ma, in close agreement with previous studies (e.g., ~112 Ma;[21] ~112.8 Ma;[23] 113 Ma[29]) (Table 3).

The cyclostratigraphic analysis of all four proxy datasets (Supplementary Fig. 5) indicates a clear imprint of $10^3$–$10^6$ yr periodicities compatible with the Milankovitch spectral peak ratio for Albian–Aptian times[36]. We also identified a pervasive, ~3.0–0.8 Myr-band above a 99.9% confidence level which is detectable for all proxy datasets, similarly to those recogniseable in sedimentary records worldwide for distinct geological intervals—the Cenozoic and Mesozoic[61,61], Early and Late Cretaceous (e.g.,

refs. [38,62]), Carboniferous (early Moscovian)[63], and Ordovician (Darriwilian)[64].

The late-Early Cretaceous was a warm period with super-greenhouse conditions, weak latitudinal temperature gradients, and an absence of ice sheets (e.g., refs. [54]). However, in the last few decades, some authors (e.g., refs. [25,58,59]) suggested that cooler interludes punctuated these conditions. Colder conditions for the Aptian–Albian were indicated by stable isotope records from Exmouth Plateau (NW Australia)[65]. A re-evaluation of calcareous nannofossil data for Sites 511, 545 (DSDP), 693A, and 766 (ODP) suggested that the late-Early Cretaceous greenhouse had a much more complex climatic history than previously postulated (e.g., ref. [25]). The timespan of the late Aptian cooling was constrained using geochemical and micropaleontological data from DSDP Site 545[8]. They suggested two cooling and warming cycles (named cycles I and II) associated with the late-Early Cretaceous "cold snap". Here, a noticeable drop in $\delta^{18}O$ (Fig. 4b) precedes at least two icehouse-linked 1.2-Myr obliquity modulations. Our results give the duration of the cold snap as recorded in the PLG core as ~2.4 Myr.

Data from this study support the dominant 405-kyr eccentricity cycles that allow us to propose a high-resolution cyclostratigraphic evaluation. From an astronomical tuning of the long-eccentricity low-pass filter output by the g2-g5 target curve from the La2004 orbital solution, it was possible: (i) to propose a floating astronomical timescale (ATS) based on 18 long-eccentricity cycles that enabled us to construct a ~7.2 Myr age model for the Aptian interval; (ii) to better constrain the timing and duration of events recorded by black-shale intervals: ~118.8 Ma /~920 kyr, ~117.9 Ma/~130 kyr, ~116.1 Ma/~150 kyr, ~113.7 Ma/~30 kyr, and ~112.9 Ma/~90 kyr for the Selli, Wezel, Fallot, 113/Jacob, and Kilian Levels, respectively (Fig. 4e); and (iii) to infer the duration of the "cold snap" based on our age model for the PLG core (~2.4 Myr) (Fig. 4e). Our data provide a new chronostratigraphic framework for Aptian times and new constraints on its related biological, chemical and geomagnetic events.

## Methods

**Geological setting and sampling strategy.** The PLG drill hole cored the uppermost Barremian–lowermost Cenomanian succession of the Umbria-Marche Basin deposited in the southern margin of the central–western Tethys Ocean[1515]. These pelagic sediments formed following the lithification of the nannofossil-planktonic foraminiferal ooze deposited well above the calcite compensation depth at middle to lower bathyal depths (1000–1500 m) and at ~20°N paleolatitude[15,32,33,45] (Fig. 1). This succession extends from the uppermost part of the Maiolica Formation (Tithonian to lower Aptian) through the entire Marne a Fucoidi Formation (lower Aptian to uppermost Albian) to the lower part of the Scaglia Bianca Formation (uppermost Albian to lowermost Cenomanian).

The uppermost Barremian–lowermost Albian succession investigated here is represented by thin-medium white to gray limestones interbedded with black shales of the Maiolica Formation and by the overlying distinctive varicolored interlude with more shale of the lower part of the Marne a Fucoidi Formation. The latter consists of thinly interbedded pale reddish to dark reddish, pale olive to dark reddish-brown, and pale olive to greyish-olive marlstones and calcareous marlstones together with dark gray to black carbon-rich shales, commonly with a low carbonate content, and yellowish-gray to light gray more or less argillaceous limestones (Fig. 2).

Some distinctive organic-rich black shale and calcareous mudstone marker beds occur within the Aptian interval, some of which have been correlated with varying degrees of success with black-shale horizons elsewhere and identified as the regional sedimentary expression of OAE 1a and OAE 1b[11,15,45]. From bottom to top, they are: (1) the Selli Level, known as the organic-rich expression of OAE 1a, (2) the Wezel Level horizon, (3) the Fallot Level, (4) the 113/Jacob Level, and (5) the Kilian Level (Fig. 2). Considering the base of M0r as the base of the Aptian, the Barremian–Aptian boundary in the PLG core falls at 95.10 m[33]. However, there is not yet a Global Boundary Stratotype Section and Point (GSSP) for this material and its definition are still ongoing. The top of the Aptian is based on the LO of the planktonic foraminifera *M. renilaevis* by definition and it is represented in GSSP for the base of the Albian stage at the Col de Pré-Guittard, in southeast France[58]. In the PLG core, this biohorizon which defines the Aptian−Albian boundary is placed at 63.40 m (Fig. 2).

Discrete ~8 cm³ cubic samples were then cut from the center of the split working halve for paleomagnetic analyses. A total amount of 1227 cubic samples were

collected along the studied portion of the PLG core (from 96.02 to 60.00 m; average sampling resolution of ~3 cm). A total of 355 paleomagnetic cubic samples were also used to measure the stable isotopes ($\delta^{18}O$ and $\delta^{13}C$) with an ~10-cm resolution.

**Rock magnetism.** Rock magnetic parameters have been largely used to identify orbital cycles in sedimentary strata[66], being fast, low-cost and non-destructive, and allowing analysis of large sample populations. Low-field magnetic susceptibility (MS, $\chi$ in $m^3/kg$) is an indirect measurement of the concentration of paramagnetic and ferromagnetic minerals in geological samples. It has been extensively used as a proxy for terrestrial detrital input in mixed carbonate-clay successions[67]. Nevertheless, variations in the MS are due to several other factors, such as magnetic mineral concentration, composition, grain size, and shape[66]. The ARM has been considered in cyclostratigraphic investigations as an alternative proxy to MS[66]. Notably, the ARM obtained at 100 mT (hereafter referred to as ARM for simplification) is useful in providing information on the fine-grained ( < 20 μm) low-coercivity ferromagnetic minerals[66]. Our rock magnetic cyclostratigraphic analyses comprise both MS and ARM datasets to provide a better assessment and comparison of the spectral content and depositional time range[63,66]. The MS measurements were carried out at the Laboratório de Paleomagnetismo of Universidade de São Paulo (USPMag). They were made on an MFK1-FA Multi-Function Kappabridge at an operating frequency of 976 Hz, in a field of 200 A/m. The acquisition of the ARM dataset was carried out by the following protocol: remanence measurements were all made in a SQUID magnetometer model 755 (2G-Enterprises), housed in a magnetically shielded room with internal field < 500 nT at USPMag. Samples underwent a stepwise alternating field (AF) demagnetization over 17 steps: 0–4 mT (step = 2 mT), 4–10 mT (step = 3 mT), 10–40 mT (step = 5 mT), 40–100 mT (step = 10 mT). After AF demagnetization, the sample was submitted to a stepwise ARM acquisition along the same AF demagnetization steps until 100 mT with a direct current bias field of 0.05 mT.

**Stable $\delta^{18}O$ and $\delta^{13}C$ isotopes.** It is well-known that specific stable isotope ratios are particularly useful for cyclostratigraphic analyses because they are independent of facies and have a global significance[67]. Geochemical parameters can be used to analyze the cycles because they are sensitive to climate and environmental changes[68]. Therefore, cyclostratigraphic analysis of $\delta^{18}O$ and $\delta^{13}C$ datasets are often used for paleoclimatic studies to detect orbitally forced patterns[38]. Carbon ($\delta^{13}C$) and oxygen ($\delta^{18}O$) isotope analyses of the carbonate fraction were performed at the University of Oxford (UO), (UK) and Research Centre of Petrobras, Brazil (CENPES). 355 bulk-rock samples every ~10 cm from the interval 60.14–95.99 m of PLG core were used in this study. A rotary drill was used in order to sample powder from the cubes used for magnetostratigraphic measurements. During sampling, veins likely containing diagenetic carbonate were avoided. A total of 6 triplicates and 28 duplicates of the same depth were used in order to evaluate the variability in isotopic determinations. The powders of 313 samples were analyzed using a VG Isogas Prism II mass spectrometer with an online VG Isocarb common acid-bath preparation system at UO. All these samples were cleaned using acetone [$(CH_3)_2CO$] and dried at 60 °C for at least 30 min. The powders of 33 samples were analyzed using a Kiel IV carbonate device coupled to Thermo Delta V Advantage mass spectrometer and nine samples were analyzed using a Gas Bench II carbonate device coupled to a Thermo Scientific Delta V mass spectrometer. Samples were reacted with purified phosphoric acid ($H_3PO_4$) at 70–90 °C in all instruments. The calibration was undertaken using the Oxford in-house Carrara marble standard (NOCZ) and NBS-19 (TS-Limestone). Data are reported relative to the Vienna Pee Dee Belemnite (VPDB) scale. The reproducibilities of replicated standards (1σ) were < 0.09‰ for $\delta^{13}C$ and < 0.10‰ for $\delta^{18}O$. The maximum difference between triplicate and duplicate samples from the same depth were 0.31‰ for $\delta^{13}C$ and 0.39 ‰ for $\delta^{18}O$. In this study, we compared these results with those provided by the rock magnetic proxies (MS and ARM) to verify a possible astronomical imprint in the PLG succession.

**Planktonic foraminifera.** The sample-set consists of 720 bulk-rock samples. At least 30 g of rock were processed for each sample using different methodologies according to the lithology and hardness of the sediment. Samples from softer lithologies were soaked in hydrogen peroxide and desogen. Where required, samples were additionally treated the surfactant benzalkonium chloride. Samples from hard lithologies were mechanically disaggregated into small fragments (3–8 mm) and treated following the cold acetolysis technique by sieving through a 40 μm mesh and drying at 50 °C. The cold acetolysis method enabled the extraction of generally easily identifiable foraminifera even from indurated limestones. This technique offered the possibility of accurate taxonomic determination and detailed analyses of planktonic foraminiferal assemblages, allowing a more precise placement of primary and secondary bioevents and zonal boundaries. Planktonic foraminifera from the washed residues were studied under a stereomicroscope to characterize assemblages and identify biostratigraphic marker species. Taxonomic concepts for genera and species of[10,11,45] are characterized were followed.

**Calcareous nannofossils.** Calcareous nannofossil assemblages were semiquantitatively investigated using a Zeiss AxioCam Imaging polarizing light microscope at 1250X magnification. The sediment was processed to obtain homogeneous smear slides following the standard preparation technique: a small chip of rock was powdered in a mortar with distillate water buffered with ammonium. Few drops of the solution were smeared on a cover glass, dried on a hot plate, and mounted on a glass slide with two drops of Norland optical adhesive. A total of 45 smear slides were analyzed; in each slide nannofossil species abundances were logged as follows: A = abundant, >1 specimen/field of view. C = Common, 1 specimen in 50/fields of view. F = Few, 1 specimen in 100/fields of view. R = Rare, 1 specimen in 200/fields of view. 300 fields of view were scanned in each slide.

**Cyclostratigraphy.** Cyclostratigraphic analyses for MS, ARM, $\delta^{13}C$, and $\delta^{18}O$ datasets were performed with the MATLAB®-based Acycle software version 2.4.1[69]. All datasets were log-transformed to harmonize fluctuation[66] before undergoing linear interpolation and uniformly resampling every 2 cm (for MS and ARM data) and 10 cm (for $\delta^{13}C$ and $\delta^{18}O$ data), which correspond to the mean spacing of the magnetic and geochemical series, respectively. The resulting datasets were then linearly detrended. Spectral analysis was carried out with the prolate multitaper spectral estimator against a robust red noise null model[70] with mean, 90%, 95%, 99%, and 99.9% confidence levels. Additionally, we performed eFFT[66] analyses to observe the transience/persistence related to the spectral frequencies along MS, ARM, $\delta^{13}C$, and $\delta^{18}O$ series, as well as to identify potential fluctuating (SARs)[66]. The short eccentricity has a mean period of ~100 kyr, with two modes of ~123 kyr and ~95 kyr[47]. The mean periods of obliquity and precession diminish back through time due to tidal friction. The astronomical solution shows the obliquity and climatic precession cycles may have exhibited main periodicities of ~38.7 kyr, ~22.91 kyr, and 18.45 kyr[36] at the Aptian–Albian boundary ~113 Ma ago[7]. The ~405-kyr long eccentricity is the expression of the g2-g5 orbital perihelia of Venus and Jupiter and, because of the large mass of the latter, it is highly stable over geological times. Hence, it has been considered in the literature as a suitable metronome in astrochronology investigations along the Phanerozoic[66]. A low-pass filter allowed us to isolate the interpreted long-eccentricity component sinusoidal curve from the datasets tuned according to the g2-g5 target curve from La2004 astronomical solution[39], which does not differ considerably from the La2011 solution for Aptian–Albian times[62]. Astronomical tunings based on the ~100 kyr short eccentricity could be unreliable for time intervals older than 50–60 Ma due to the chaotic evolution of the Solar System[39,63] and hence it was not considered here for astronomical tuning.

In order to find the optimal sedimentation rate based on our astrochronological interpretation for the PLG core, we performed correlation coefficient (COCO) and its evolutionary variant (eCOCO) analyses[40] with 5000 Monte Carlo simulations on tested sedimentation rates ranging from 0.05 to 3.00 cm/kyr, after removing different long-term trends with a 'lowess' smoother (sliding windows of 4, 6, and 8 m) (Supplementary Material).

## Data availability
The MS, ARM, $\delta^{13}C$, and $\delta^{18}O$ datasets used in this study are available in the Zenodo repository https://doi.org/10.5281/zenodo.6383426.

## Code availability
All mathematical treatments were performed with the freeware Acycle designed by ref. [69] and made publicly available by these authors at https://github.com/minsongli/acycle.

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

## Acknowledgements

The Poggio le Guaine core is an integral part of the FUSP (Fundação de Apoio à Universidade de São Paulo)-Petrobras 2405 project. The paper is an integral part of the Project: Processamento e interpretação de dados magnetoestratigráficos do Cretáceo das Bacias Brasileiras, which is financially supported by Petróleo Brasileiro S.A.—Petrobras (FAURGS 8368). The authors are grateful to Professors Hugh C. Kenkyns and Stuart Robinson (University of Oxford, U.K.) for the valuable discussions that contributed to this study. We all are grateful to São Paulo State Research Foundation (FAPESP) to academically support this work through the CORE project (16/24946-9). Support for this work came from the National Council for Scientific and Technological Development (CNPq—grants 141093/2018-8 to C.G.L., 201508/2009-5, 427280/2018-4 and 311231/2021-7 to J.F.S., 132076/2019-5 to M.V.L.K., 141093/2018-8 to D.R.F., 312453/2019-1 to L. Janikian, 303990/2018-0 to R.P.A.) the Research Support Foundation of the State of Rio Grande do Sul (FAPERGS—grant 16/2551-0000213-4 to J.F.S.) the Foundation Carlos Chagas Filho Research Support of the State of Rio de Janeiro (FAPERJ—grant E-26/203.302/2017 to D.R.F.) the Coordination for the Improvement of Higher Education Personnel (CAPES—grant 313253/2017-0 to D.R.F.). We acknowledge the efficient editorial handling of the manuscript. Criticism by reviewers led to significant improvement of the article.

## Author contributions

C.G.L. and J.F.S. conceived the study. J.F.S, R.C., F.F., L. Jovane, M.F., L.Janikian, R.P.A., and R.I.F.T. conducted the fieldwork. C.G.L., M.V.L.K., and D.R.F. conducted the time series analysis. J.F. and C.G.L. conducted rock magnetic experiments. L.R.T. conducted stable isotopes analyses. R.C., F.F., and S.G. conducted the biostratigraphy analyses. All authors contributed to the manuscript preparation, interpretation, discussion, and writing, led by C.G.L.

## Competing interests

The authors declare no competing interests.
