## [Peer Review File · Nature Communications]

Astronomical tuning of the Aptian Stage and its implications for age recalibrations and paleoclimatic eventsReviewers' Comments:

Reviewer #1:

Remarks to the Author:

The geochronology of the Aptian Stage is highly disputed. This manuscript evaluates multiple datasets from the Poggio le Guanine core in the Umbria-Marche Basin of Italy. The conclusions sound interesting, but some of the important ones suffer from problems.

1. It claims the chaotic transition from 2.4 myr to 1.2 myr at ca. 115.5 Ma. The evidence is very weak and purely based on the subjective recognition based on the low pass filtered output. And this claimed event doesn't have any theoretical support. Perhaps no existing astronomical solution is able to produce the claimed events, although it is still possible that none of the solutions is able to catch this event due to the chaos of the Solar system. In sum, this conclusion, without further evidence and robust evaluation, is not sufficient for *Nat Commun*, a top-tier journal in geosciences.

2. It also argues a time span of 7.1 myr duration for the Aptian. This duration is much shorter than that published by Huang et al., 2010, but generally similar to other recent calculations. However, data from the low-sedimentation interval – Selli – is incomplete. Thus the long eccentricity cycles marked as E13 to E17 are arbitrarily inferred, thus is not able to provide constraints for the Aptian duration. Therefore the third conclusion in the abstract cannot stand either.

3. The writing style of the abstract needs improvement. There is no background, no raised questions, nor no test hypothesis. Moreover, this Aptian cyclostratigraphy work cannot impact the geologic time scale from the Jurassic to the Early Cretaceous because the Aptian duration is not so long.

Taken together, this manuscript doesn't reach the quality for nature communications. However, it does have its merits and worth publishing in a good geology journal.

Reviewer #2:

Remarks to the Author:

The major claims of the paper are a new age estimate for the Barremian/Aptian boundary and new estimates of the timing and duration of key events of the Aptian including OAEs and cooling/warming cycles. The estimate for the B/A boundary is not entirely novel as very similar estimates were used in the 2020 GTS. But I think the community would appreciate the independent validation of the updated stage boundary age estimate presented in this study. Similarly, the estimates of timing and duration of Aptian events provide independent validation of competing age dates from previous studies, which is novel in that this method can mostly reproduce the ages produced from radiometric methods. Notably, they find the duration and timing of the late Aptian 'cold snap' to be consistent with other estimates of the timing of the chaotic transition and the duration of the event as constrained by bio- and chemo-stratigraphic studies. This is of broad interest since it addresses emerging questions about the extent and degree of global warmth during the mid-Cretaceous and suggests a relationship between the cold snap and orbital parameters. These results are of an immediate interest to different geoscience subfields. This paper will influence thinking in the field because it demonstrates that astronomically tuned age models can reproduce radiometrically obtained dates, but it also furthers the use of these kinds of age models without consideration of sequence stratigraphy, which is a looming issue with the method.

A major critique of the work centers around the disregard for sequence stratigraphy when interpreting the records of the core and especially when relating the study core to other localities. Sequence boundaries are a major problem for the application of astronomically tuned age models but could be first order detected through an evaluation of the impact of diagenesis on the isotope records. A scatterplot of oxygen and carbon data would help the reader assess the potential role of diagenesis in the records which could in turn help with the identification of sequence boundaries and flooding

surfaces.

The study provides constraints on the duration and age of the Selli horizon (and other black shale horizons), but the authors do not discuss why sometimes they can replicate results from previous studies but at other times they are not be able to replicate the results. Is this sampling bias, preservation, or hiatus? A discussion of this would strengthen their arguments about fidelity in their age model. Another claim the authors make is that ages of OJP and HALIP suggest these incidents of volcanism are triggers of OAE1a. This claim is not well supported because there is no discussion on the age dates of HALIP nor is there a discussion on what defines OAE1a (is it the excursion, if so what part, or is it the bottom of the Selli level?). If the event is defined by different stratigraphic horizons at different localities then a discussion of how the authors define OAE1a is critical.

The work can be reproduced, but more information on the definition of particular stratigraphic horizons should be provided. Specifically, the paragraph made up of lines 454-469 is critically important and should be worked into the main text (perhaps the introduction) because it provides the reader with important information for evaluating the stratigraphic correlations among cores. Figure 2 should have information about the isotope zones names since they are referenced liberally throughout the text. It should be clearly stated if the carbon isotope data is from carbonate or organic matter.

Author's Response to the Review Comments

Manuscript #: NCOMMS-21-18867

Title of Paper: Astronomical tuning of the Aptian Stage: implications for age recalibrations and paleoclimatic events

Authors: C. G. Leandro, J. F. Savian, M. V. L. Kochhann, D. R. Franco, R. Coccioni, F. Frontalini, S. Gardin, L. Jovane, M. Figueiredo, L. R. Tedeschi, L. Janikian, R. P. Almeida & R. I. F. Trindade

Date Sent: November 10, 2021

Response to Comments from Reviewer 1

General Comments:

The geochronology of the Aptian State is highly disputed. This manuscript evaluates multiple datasets from the Poggio le Guaine core in the Umbria-Marche Basin of Italy. The conclusions sound interesting, but some of the important ones suffer from problems.

Response:

We fully agree with the R1 that the geochronology of the Aptian Stage is highly disputed in the literature.

The studied interval includes the Aptian oceanic anoxic events (OAEs). In the early Aptian, the OAE 1a, which is one of the most extreme examples of global anoxia in the mid-Cretaceous greenhouse world. As you pointed out, we present multiple datasets from the Poggio le Guaine (PLG) core, where the most continuous, complete, and best preserved Aptian-Albian succession is exposed in the Umbria-Marche Basin (UMB) of the northern Apennines of central Italy. We greatly appreciate the reviewer's efforts to carefully review the paper and the valuable suggestions offered. Following the recommendations, we will clarify point-by-point the questions indicated by the reviewer.

Comment 1:

It claims the chaotic transition from 2.4 myr to 1.2 myr at ca. 115.5 Ma. The evidence is very weak and purely based on the subjective recognition based on the low pass filtered output. And this claimed event doesn't have any theoretical support. Perhaps no existing astronomical solution is able to produce the claimed events, although it is still possible that

none of the solutions is able to catch this event due to the chaos of the Solar system. In sum, this conclusion, without further evidence and robust evaluation, is not sufficient for Nat Commun, a top-tier journal in geosciences.

Response:

The authors agree with reviewer 1 about the Comment 1, especially regarding the impossibility of an unequivocal inference of the $\sim 2.4\text{-Myr}/1.2\text{-Myr}$ transition at $\sim 115.5\text{ Ma}$, the more that the chronostratigraphic framework provided for the PLG core is not supported by radioisotopic dating. Therefore, we completely removed such inference from the new version of our manuscript. Nevertheless, we would like to discuss some of the statements contained in this comment, on the basis of the following points:

1) There are a number of recent studies which pointed out that secular resonance transitions, resulting from the chaotic dynamical behaviour of the Solar System, may have had some implications in paleoclimatic events throughout the geological time. The first geological evidence for a geochronologically well constrained chaotic resonance transition during Cretaceous times was presented in a letter published in *Nature* (Ma et al., *Nature* 542, 468-470 (2017)). In this study, the authors indicated that a chaotic transition may have taken place during the OAE 3 ($\sim 84\text{-}88\text{ Myr}$), by means of an integrated radioisotopic and astronomical timescale for North American Cretaceous records. Furthermore, they also stated that "... numerous studies have proposed mechanistic linkages between palaeoclimate events and 'nodes' in the theoretical astronomical-insolation solutions; examples include the pacing of Antarctic ice-sheet growth during the Oligocene (Pälike et al., 2006) and the timing of Cretaceous Oceanic Anoxic Event 2 (Mitchell et al., 2008)". The existence of a 2:1 resonance of the $\sim 2.4\text{-}$ and $\sim 1.2\text{-Myr}$ 'grand' cycles has also been suggested for Early Paleozoic times (Crampton et al., 2018). For more recent times, Westerhold et al. (2017) suggested the occurrence of a similar chaotic transition at $\sim 52\text{ Ma}$, from a bio- and magnetostratigraphically calibrated stable isotope compilation carried out from collected cores of ODP Sites 1258 (Leg 207, Demerara Rise), 1262, 1263, 1265, and 1267 (South Atlantic Ocean). Therefore, as also advocated by Ma et al. (2017), there are evidences discussed in the literature on the $\sim 2.4/1.2\text{ Myr}$ resonance transition, at distinctive geological intervals, which point out for its origin in a chaotic dynamical behaviour of the Solar System, with a potential impact for future improvements in theoretical astronomical-insolation solutions.

2) It is also important to highlight that million-year-scale signals were clearly evidenced by the spectral analysis of the PLG core, as presented in our manuscript. As summarized below in Figure 1, a pervasive ~ 3.0 - 0.8 Myr-band is detectable for the four analyzed proxy datasets (MS, ARM, $\delta^{13}\text{C}$ and $\delta^{18}\text{O}$), all of them above the confidence level (99.9%). Therefore, according to this evidence, as well as our agreement to discard the inference on the ~ 2.4 -Myr/ 1.2 -Myr transition, we propose to change the acronyms EAM (~ 2.4 Myr) and OAM (~ 1.2 Myr) by MSB (million-year scale band) in the Figure 3 and Supplementary Fig. 2 of the new version of our manuscript.

Fig. 1 Spectral analysis of PLG core data. 2π multitaper power spectra for **a** magnetic susceptibility (MS); **b** anhysteretic remanent magnetization (ARM); **c** $\delta^{13}\text{C}$; **d** $\delta^{18}\text{O}$ data, with the AR(1) red noise spectral model and 85%, 90%, 95%, and 99% confidence levels (c.l.) for null hypothesis testing. The rectangle (dashed line) indicates the frequency range for the MSB (million-year scale band) - which was referred to "EAM/OAM" in Figure 3 and Supplementary Fig. 1 of the original version of our manuscript.

Comment 2:

It also argues a time span of 7.1 myr duration for the Aptian. This duration is much shorter than that published by Huang et al., 2010, but generally similar to other recent calculations. However, data from the low-sedimentation interval – Selli – is incomplete. Thus the long eccentricity cycles marked as E13 to E17 are arbitrarily inferred, thus is not able to provide constraints for the Aptian duration. Therefore the third conclusion in the abstract cannot stand either.

Response:

We probably were not clear about the steps that guided us towards our proposition of the astronomical timescale (ATS) for the PLG core. The indication of the E13 - E17 long eccentricity range inferred for the Selli Level was not made arbitrarily.

We followed these steps: (1) due to the lack of radiometric ages for the Marne a Fucoidi Formation, we anchored our astronomical tuning to the top of the Kilian black shale (~112.8 Ma; Sabatino et al., 2018); (2) a careful compilation (disposed at Tables 1 and 3 in the manuscript) shows the time spans and ages suggested for Aptian black shale levels; (3) we adopted the estimated time span for the Selli Level provided by Malinverno et al. (2010), based on $\delta^{13}\text{C}$ data of the Cismon APTICORE borehole (1.11 ± 0.11 Myr - see Fig. 5B of the manuscript); and (4) based on the assumptions mentioned before, we built our ~405-kyr tuned age model for the PLG core based on the long eccentricity lowpass filter output from MS data (Fig. 4B in the manuscript) and the g2-g5 target curve from the La2004 solution for the Aptian-Albian interval (Fig. 4C in the manuscript).

Particularly regarding the Selli Level, it was not possible to directly estimate its mean duration from our cyclostratigraphic work, as the spectral peaks were badly resolved and the orbital-scale harmonic content was not “traceable” within this interval. Therefore, in order to avoid cyclostratigraphic misinterpretations, we chose to remove the data in this stratigraphic interval (DL = 195 cm - from 91.19 to 89.24 m) and assumed the duration for the Selli Level as proposed by Malinverno et al. (2010; 1.11 ± 0.11 Myr). It is noteworthy that even if we use the longest duration proposed for the Selli (~1.4 Myr; Huang et al., 2010), the estimated time interval for the Aptian would be ~7.6 Myr, still 5.3 Myr shorter than the duration published by Huang et al. (2010). After the astronomical tuning was performed, our proposition of a cyclostratigraphic age model for the PLG core led to an indirect estimate

for the duration of the Selli Level of ~920 kyr, which is strikingly compatible to the Malinverno et al. (2010) proposition (1.11 ± 0.11 Myr). Furthermore, there are other strong indications that clearly point out for the high quality of our cyclostratigraphic age model:

(i) Our estimated age of ~120.2 Ma for the Aptian-Barremian / base M0r boundary (Fig. 5C of the manuscript) is compatible with a ^{40}Ar - ^{39}Ar radiometric age (121.2 ± 0.5 Ma) obtained for early M0r based on lava flows from the Mashenmiao-Zhuanchengzi (MZ) section, NE China (He et al, 2008), as well as with the two most recent studies that discussed the age for the Aptian-Barremian boundary: (a) 121-122 Ma for the U-Pb zircon dating of Barremian tephra layers (Midtkandal et al., 2016); and (b) 121.2 ± 0.4 Ma for the U-Pb dating of the bentonite bedrock (DH1 core) of Svalbard, Norway (Zhang et al., 2021), which is currently used as a reference for the Aptian-Barremian boundary in GTS 2020;

(ii) According to Kennedy et al. (2017), the Aptian-Albian boundary is defined at the lowest occurrence of the planktonic foraminifera *Microhedbergella renilaevis* that is placed at ~ 63.40 m for the PLG core. Our age model provides an age of ~ 113.1 Ma for this stratigraphic height, which is remarkably compatible with the age for the Aptian-Albian boundary as defined by the Geological Time Scale (GTS) 2020 (113.2 ± 0.3 Ma; Gradstein et. al., 2020). Following the publication of GTS 2020, we use this radiometric age (~113.1) to tie our age model.

Another important observation is that if we had opted for constraining our age model to the top of the Selli Level (~118.4 Ma) and thereafter assume the duration proposed by Malinverno et al. (2010), where the Barremian-Aptian boundary would be placed at ~120.3 Ma, which is virtually the same age proposed by our age model (~120.2 Ma). Therefore, we are confident that our interpretation for the Selli Level duration is coherent, as well as that our cyclostratigraphic age model is consistent and provides a reliable chronostratigraphic interpretation for the entire PLG stratigraphy.

Comment 3:

The writing style of the abstract needs improvement. There is no background, no raised questions, nor no test hypothesis. Moreover, this Aptian cyclostratigraphy work cannot impact the geologic time scale from the Jurassic to the Early Cretaceous because the Aptian duration is not so long.

Response:

Thank you for your comments. We have rewritten the abstract as per the suggestions and clarified what we wanted to express. A change in the Aptian duration will consequently influence the ages of the pre-Cretaceous periods. Nevertheless, we changed the paragraph accordingly to make it unambiguous.

L. 33-43: *“The Aptian was characterized by dramatic tectonic, oceanographic, climatic and biotic changes and its record is punctuated by Oceanic Anoxic Events (OAEs). The timing and duration of these events are still contentious, particularly the age of the Barremian-Aptian boundary. This study presents a cyclostratigraphic evaluation of a high-resolution multiproxy dataset ($\delta^{13}\text{C}$, $\delta^{18}\text{O}$, MS and ARM) from the Poggio le Guaine core. The identification of Milankovitch-band imprints allowed us to construct a 405-kyr astronomically-tuned age model that provides new constraints for the Aptian climato-chronostratigraphic framework. Based on the astronomical tuning, we propose: (i) a timespan of ~ 7.1 Myr for the Aptian; (ii) a timespan of ~ 420 kyr for the magnetic polarity Chron M0r and an age of ~ 120.2 Ma for the Barremian–Aptian boundary; and (iii) new age constraints on the onset and duration of Aptian OAEs and the ‘cold snap’. The new framework significantly impacts the Early Cretaceous geological timescale.”*

Comment 4:

Taken together, this manuscript doesn't reach the quality for nature communications. However, it does have its merits and worth publishing in a good geology journal.

Response:

Based on our answers detailed above, we kindly ask for the reviewer reconsideration. We believe our study has the potential to be published in *Nature Communications*, because we present an effective dating technique based on four proxies that can fill the shortage of radiometric dating in the sedimentary records. We show an age for the Barremian-Aptian boundary very similar to what was shown in the GTS 2020, and we estimate durations of major Aptian anoxic events, including OAEs and cooling/warming cycles. Our results are of

interest to different subfields of geosciences as we address emerging questions about the extent and degree of global warming during the Middle Cretaceous. The time and duration of Middle Cretaceous hyperthermals are excellent analogs to the present climate changes, which provide a conclusion of interest to an interdisciplinary readership. Major advances in our understanding of paleoclimate change can derive from a precise reconstruction of the periods, amplitudes and phases of the 'Milankovitch cycles 'of precession, obliquity and eccentricity. Apart from paleoclimatic implications, our results include testing and calibrating a theoretical astronomical solution (La2004), and refining the chronologies of the deep past.

Response to Comments from Reviewer #2

General Comments:

The major claims of the paper are a new age estimate for the Barremian/Aptian boundary and new estimates of the timing and duration of key events of the Aptian including OAEs and cooling/warming cycles. The estimate for the B/A boundary is not entirely novel as very similar estimates were used in the 2020 GTS. But I think the community would appreciate the independent validation of the updated stage boundary age estimate presented in this study. Similarly, the estimates of timing and duration of Aptian events provide independent validation of competing age dates from previous studies, which is novel in that this method can mostly reproduce the ages produced from radiometric methods. Notably, they find the duration and timing of the late Aptian 'cold snap 'to be consistent with other estimates of the timing of the chaotic transition and the duration of the event as constrained by bio- and chemo-stratigraphic studies. This is of broad interest since it addresses emerging questions about the extent and degree of global warmth during the mid-Cretaceous and suggests a relationship between the cold snap and orbital parameters. These results are of an immediate interest to different geoscience subfields. This paper will influence thinking in the field because it demonstrates that astronomically tuned age models can reproduce radiometrically obtained dates, but it also furthers the use of these kinds of age models without consideration of sequence stratigraphy, which is a looming issue with the method.

Response:

We thank the reviewer for the extremely positive comments on this manuscript, and have done our utmost to address the points that have been raised.

Comment 1:

A major critique of the work centers around the disregard for sequence stratigraphy when interpreting the records of the core and especially when relating the study core to other localities. Sequence boundaries are a major problem for the application of astronomically tuned age models but could be first order detected through an evaluation of the impact of diagenesis on the isotope records. A scatterplot of oxygen and carbon data would help the reader assess the potential role of diagenesis in the records which could in turn help with the identification of sequence boundaries and flooding surfaces.

Response:

We followed the reviewer's suggestion and made the C-O isotopes biplots (see below). It shows no clear correlation ($R = 0.0084$), attesting no significant diagenetic imprints on the isotope signal. The sampled area corresponds to the distal sector of the Umbria-Marche basin. This sector of the basin is typically characterized by almost continuous slow deposition, where sequence boundaries correspond to conformable surfaces within a complete pelagic succession.

The high-resolution biostratigraphic record in the studied section also suggests that the sedimentation in this sector is roughly continuous through time in the interval of interest and indicates about 1000 to 1500 meters of paleobathymetry. Thus, the core would not be in the context of influence by diagenetic variations due to sea level fluctuation. Owing to the continuous deposition in a pelagic setting, a reduced tectonic overprint, and excellent age control through magnetostratigraphy, biostratigraphy, chemostratigraphy and tephrostratigraphy, these sections are internationally recognized as reference record for part of the Cretaceous time scale.

We have added a detailed discussion in the main text: *“In our PLG section, diagenetic overprint probably has taken place on all section in order to turn $\delta^{18}\text{O}$ profile noisy and $\delta^{18}\text{O}$ values relatively depleted than a pristine fossil record. Nevertheless, the relatively higher $\delta^{18}\text{O}$ values in the interval in the interval 74.10–63.35 m matches the same stratigraphic correlation observed in Piobiccio core (Italy) and DSDP Site 463 (Pacific Realm)⁷ based on bio- and carbon isotope stratigraphy, where nannofossil proxies together with relatively higher $\delta^{18}\text{O}$ values suggest relatively colder temperatures during C9 to C11 segment in the late Aptian. Moreover, it also shows good agreement to DSDP Site 545⁸ based on the same bio- and carbon isotope stratigraphy, where cold snap has been identified based on TEX₈₆ proxies, which are specific organic compounds. Therefore, the relatively higher $\delta^{18}\text{O}$ values*

in our section has been interpreted as a 'cold snap' record." (L. 174–183)

Fig. 2 $\delta^{18}\text{O}$ and $\delta^{13}\text{C}$ cross-plot of carbonate samples from PLG Core.

Comment 2:

The study provides constraints on the duration and age of the Selli horizon (and other black shale horizons), but the authors do not discuss why sometimes they can replicate results from previous studies but at other times they are not be able to replicate the results. Is this sampling bias, preservation, or hiatus? A discussion of this would strengthen their arguments about fidelity in their age model.

Response:

We thank the Reviewer for the comment. There is only one study that estimates the timespan for all black shale levels of the Aptian stage and it shows a difference of ~5 Myr for the Chron M0r base (Huang et al., 2010). This is the main reason why we opted not to assume the same durations. For instance, Huang et al. (2010) estimated that the Fallot Level has lasted ~360 kyr, while we obtained a timespan of less than a half of that (~150 kyr). We discussed in detail about the Fallot Level in:

(L. 377–390) *"We identify a sustained peak in stable isotopes in the PLG core between 67.52 and 67.44 m, interpreted as the so-called Fallot Level. This level is characterised by a series of black shale layers formed under different forcing mechanisms, including enhanced burial*

of organic matter due to eutrophic conditions or low oxygenation at the seafloor²¹. According to some authors in the Tethyan realm (e.g., refs. ^{21,53}), four prominent black shales (FA 2', 2'', 3 and 4) are recognisable.

The black shale FA3 is 8 cm thick in the PLG core and occurs in the Globigerinerelloides algerianus planktonic foraminiferal and NC7 calcareous nannofossil Zones (Figs. 2 and 4). The timespan for this Fallot Level black shale is ~150-kyr (Table 1), and its estimated age is ~116.1 Ma (Table 3). Notably, a timespan of ~360 kyr at ~117.8 Ma was estimated for a ~1 m thick interval in the Hedbergella trocoidea planktonic foraminiferal and the NC7 calcareous nannofossil Zones in the Piobbico core (central Italy), and it has been attributed to the Fallot Level²¹. However, black shale layers are absent in the Globigerinelloides ferreolensis, G. algerianus and H. trocoidea planktonic foraminiferal Zones in the Piobbico core. Therefore, the presence of the Fallot Level in the Piobbico core is uncertain.”

We also adopted the timescale of Malinverno et al. (2012) because Aptian ages are consistent with the radiometric age of magnetochron CM0 (He et al., 2008). This is corroborated by the GTS 2020 and the radiometric age of 121.2 ±0.4 Ma (Zhang et al., 2021). The Re-Os age of 120.4 ±3.4 Ma for the base of the Selli Level (Bottini et al., 2012) also reinforces the robustness of our age model. In this way, we have robust time constraints that tie the top and the base of the record, which allow a reliable determination of the Ocean Anoxic Events during the Aptian.

Comment 3:

Another claim the authors make is that ages of OJP and HALIP suggest these incidents of volcanism are triggers of OAE1a. This claim is not well supported because there is no discussion on the age dates of HALIP nor is there a discussion on what defines OAE1a (is it the excursion, if so what part, or is it the bottom of the Selli level?). If the event is defined by different stratigraphic horizons at different localities then a discussion of how the authors define OAE1a is critical.

Response:

The early Aptian OAE 1a is the most prominent mid-Cretaceous Oceanic Anoxic Event, typified by worldwide deposition of thick organic-rich horizons. As mid-Cretaceous OAEs were often accompanied by intensive marine biotic crises, understanding the factors that triggered the OAEs is important for unraveling the evolution of the Cretaceous marine ecosystem. Massive volcanic events associated with the formation of large basaltic plateaus

called large igneous provinces (LIPs), are the most probable triggering factors of environmental perturbations. This has been supported by radiometric ages of the basaltic plateaus corresponding to the sedimentary ages of major OAEs and the species turnovers of marine calcareous plankton.

As stated by Percival et al. (2021), we cannot completely rule out a contribution of High Arctic LIP (HALIP) related carbon in initiating OAE 1a. Nonetheless, other studies suggest that heating of organic-rich lithologies by LIP magmas likely released Hg as well as carbon species (Percival et al., 2015, 2017; Svensen et al., 2018; Jones et al., 2019; Shen et al., 2019). Thus, the lack of a Hg-cycle perturbation makes it likely that if there was any role played by the HALIP in the onset of OAE 1a, it was very minor compared to the Ontong Java Plateau (OJP).

We discussed about the link between HALIP, OJP and OAE1a in: *“The High Arctic Large Igneous Province (HALIP) has recently been postulated as an alternative carbon source, producing CO₂ by metamorphism of organic matter (estimated at ~19,200 Gt of carbon). This source requires a lower emissions volume to cause the recorded negative $\delta^{13}\text{C}$ excursion. The rapid release of aureole greenhouse gases (methane) from HALIP may have contributed to the negative $\delta^{13}\text{C}$ excursion in OAE 1a from 120.2 ± 1.9 Ma to 124.7 ± 0.3 Ma⁵². However, the Hg-cycle perturbation by the HALIP in the onset of OAE 1a was smaller compared to the OJP⁶. A third possible source is the release of methane hydrates, a suggested trigger of the negative $\delta^{13}\text{C}$ excursions from other OAEs, such as the Toarcian OAE. Based on our age estimate, the causal link between the onset of OAE 1a (C3 segment) and ages from OJP and HALIP suggests that these magmatic events helped to trigger the oceanic anoxia recorded in the Selli Level^{6,52}. As the isotopic record of the C3 segment in PLG core do not show well constrained boundaries to the lack of carbonate fraction, we were not able to add further discussion about casual links between OAE 1a and possible sources of depleted CO₂.”* (L. 357–369)

The definition of the stratigraphic position of OAE1 in the PLG core is in:

(L. 119–124) *“The Selli Level (91.19–89.24 m) occurs at the base of the PLG core (C3 to C6 or Ap3 to Ap6)^{32,33}. The $\delta^{13}\text{C}$ negative excursion in C3 (1.47‰) followed by a positive excursion (up to 4.44‰) unambiguously indicates OAE 1a. However, the PLG core is virtually carbonate-free in this interval, and the sparse $\delta^{13}\text{C}$ data do not allow us to define the two positive excursions (C4 to C6 and Ap4 to Ap6 segments) defined globally^{7,33}.”*

Comment 4:

The work can be reproduced, but more information on the definition of particular stratigraphic horizons should be provided. Specifically, the paragraph made up of lines 454-469 is critically important and should be worked into the main text (perhaps the introduction) because it provides the reader with important information for evaluating the stratigraphic correlations among cores.

Response:

We added more information on the definition about organic-rich black shales into the main text “*In addition to the Selli, others distinctive organic-rich black shale and calcareous mudstone marker beds occur within the Aptian interval, recognized mainly in the Tethyan realm. From bottom to top, they are: (1) the Wezel Level horizon²⁰ (2) the Fallot Level (~ 117.8 Ma^{21,22}), (3) the 113/Jacob Level (~ 113-115 Ma^{21,23}) and (4) the Kilian Level (~ 112-113 Ma), the last one marking the Aptian–Albian boundary^{7,11,21,24}. (Fig. 2). Although the definition of black shale levels corresponding to the OAE 1b differs, depending on the study, the Jacob and Kilian Levels are commonly accepted as records of organic-rich expressions of the first two sub-events of OAE 1b²⁵.*” (L. 70–76).

Comment 5:

Figure 2 should have information about the isotope zones names since they are referenced liberally throughout the text. It should be clearly stated if the carbon isotope data is from carbonate or organic matter.

Response:

We thank the Reviewer for the suggestion. We modified the figure according to your suggestions, including the isotope zones names, stated that the carbon isotope data is from carbonate and modified the text according to the nomenclatures used for the isotopic zones “*The $\delta^{13}\text{C}$ values vary significantly in the Aptian (1.47–4.82‰; Fig. 2c). The 95.99 m (base)–91.19 m interval displays $\delta^{13}\text{C}$ values between 1.97 and 3.67‰. The 95.12 m (Barremian–Aptian boundary) to 91.19 m corresponding to the C1, C2 and Ap1, Ap2 segments^{7,33}. The Selli Level (91.19–89.24 m) occurs at the base of the PLG core (C3 to C6 or Ap3 to Ap6)³³. The $\delta^{13}\text{C}$ negative excursion in C3 (1.47‰) followed by a positive excursion (up to 4.44‰) unambiguously indicates OAE 1a. However, the PLG core is virtually carbonate-free in this interval, and the sparse $\delta^{13}\text{C}$ data do not allow us to define the two positive excursions (C4 to C6 and Ap4 to Ap6 segments) defined globally^{7,33}. The C7/Ap7 segment⁷ corresponds to*

higher $\delta^{13}\text{C}$ values (3.65–4.82‰) at 89.24–86.12 m. The Wezel Level²⁰ (88.20–88.00 m) is marked by a slight $\delta^{13}\text{C}$ negative excursion from 4.33‰ to 3.65‰ within the C7/Ap7 segment. The intervals from 89.24–88.20 m and 88.00–86.10 m in the C7/Ap7 segment are a pale greyish-olive colour; the feature persists up to 82.10 m. From 86.12 to 78.18 m, $\delta^{13}\text{C}$ values decrease from > 4‰ to 2.85‰, characterising the C8/Ap8 (86.12–83.51 m) and C8/Ap9-Ap11 (83.36–78.18 m) segments⁷. Pale to dark reddish-brown limestones and clay marls are the predominant lithology in the interval 82.10–63.02 m. The Fallot Level occurs in the 79.37–79.31 interval within C8/Ap9–Ap11 segments⁷. The C9/Ap12 segment in the interval 78.18–73.00 m shows a characteristic increase in $\delta^{13}\text{C}$ values from 2.81‰ to 4.10‰ and defines the onset of a broad positive carbon isotope excursion in the late Aptian, which continues in the C10/Ap13-Ap15 segment. $\delta^{13}\text{C}$ values are mostly over 3.5‰ in the interval 73.00–67.70 m. A slightly increasing $\delta^{13}\text{C}$ trend culminates in a prominent peak shift (3.28‰) at 67.70 m. The 113/Jacob Level (67.44–67.52 m) is marked by a positive peak at 4.16‰ subsequently decreasing to 3.26‰ within the C10/Ap15 segment⁷. The C11/Ap16-Ap18 segment (65.42–62.26 m) shows $\delta^{13}\text{C}$ values decreasing from ~4 to ~3.39 ‰, with rapid increasing $\delta^{13}\text{C}$ values and a positive shift to 4.26‰ at the top of the black shale Kilian Level (62.64 m). The stratigraphically higher (63.02–60.14 m) section comprises pale olive to greyish-olive marls, clayey marl, marly clay and clay. Here, $\delta^{13}\text{C}$ values vary slightly from 2.82 to 3.55‰ and correspond to the C11/All segments⁷.” (L. 117–141)

Fig. 3 Integrated stratigraphy of the studied interval at PLG core. Stratigraphic framework of the PLG core with real stratigraphic depths according to bed dip measurements. Depths for the upper boundaries of the planktonic foraminiferal and calcareous nannofossil zones, and nannoconid decline and crisis biohorizons identified in the PLG core^{11,32} and this work. Changes in **a** magnetic susceptibility (MS); **b** anhysteretic remanent magnetization (ARM); **c** $\delta^{13}\text{C}$; **d** $\delta^{18}\text{O}$. The gray bands highlight the Selli, Wezel, Fallot, 113/Jacob and Kilian Levels. Codes for C-isotope segments^{7,33}.

Reviewers' Comments:

Reviewer #1:

Remarks to the Author:

I am glad to see the very detailed response to comments from both reviewers. The response clearly addressed my concerns on the long eccentricity cycles and the duration of the Aptian.

It is nice that the 2.4 myr – 1.2 myr transition has been replaced by the "million-year scale band" because the previous argument needs a lot of solid evidence that is beyond the ability of this paper.

Here is another concern: this paper only used traditional cyclostratigraphic approaches on the detection of the long eccentricity cycles based on the cycle-ratio method and bandpass filtering, which can be subjective. Recent advances in cyclostratigraphy include statistical tuning approaches, such as ASM, COCO, timeOpt, and the evolutive version of these methods. Can you do a similar analysis and show the sedimentation rate map from both conventional methods (cycle-counting) and sliding window statistical tuning approaches (either eASM, eCOCO, or eTimeOpt). At least one analysis will definitely greatly enhance our confidence in the interpretation of this nice study.

Moreover, eFFT should be explained when it is first used in the paper.

Reviewer #2:

Remarks to the Author:

This manuscript presents multi-proxy records from a core that spans a geochronologically controversial time interval, and uses astronomical tuning to address geochronological uncertainties especially as they pertain to stage boundaries and ancient climatic events. The results generated new age and duration estimates that compliment previously published dates that used radiometric methods.

The methodology, data analysis, interpretation, claims and conclusions appear sound. Although, most major conclusions are tied to geochronology, they should be of interest to a wide range of earth scientists since they concern the geologic timescale.

Author's Response to the Review Comments

Manuscript #: NCOMMS-21-18867A-Z

Title of Paper: Astronomical tuning of the Aptian Stage: implications for age recalibrations and paleoclimatic events

Authors: C. G. Leandro, J. F. Savian, M. V. L. Kochhann, D. R. Franco, R. Coccioni, F. Frontalini, S. Gardin, L. Jovane, M. Figueiredo, L. R. Tedeschi, L. Janikian, R. P. Almeida & R. I. F. Trindade

Date Sent: March 4, 2022

Response to Comments from Reviewer 1

General Comments:

I am glad to see the very detailed response to comments from both reviewers. The response clearly addressed my concerns on the long eccentricity cycles and the duration of the Aptian. It is nice that the 2.4 myr – 1.2 myr transition has been replaced by the "million-year scale band" because the previous argument needs a lot of solid evidence that is beyond the ability of this paper.

Response:

We are glad to know that we properly addressed this problem.

Comment 1:

Here is another concern: this paper only used traditional cyclostratigraphic approaches on the detection of the long eccentricity cycles based on the cycle-ratio method and bandpass filtering, which can be subjective. Recent advances in cyclostratigraphy include statistical tuning approaches, such as ASM, COCO, timeOpt, and the evolutive version of these methods. Can you do a similar analysis and show the sedimentation rate map from both conventional methods (cycle-counting) and sliding window statistical tuning approaches (either eASM, eCOCO, or eTimeOpt). At least one analysis will definitely greatly enhance our confidence in the interpretation of this nice study.

Response:

The authors are thankful for these very constructive suggestions. We agree that detection of the long eccentricity cycles based only on the cycle-ratio method and bandpass filtering can be subjective. Therefore, we provide in the newer version of the manuscript statistical tests based on the correlation coefficient (COCO) and its evolutionary variant (eCOCO) methods. For the COCO/eCOCO tests, we aimed as follows: (i) to verify the optimal sediment accumulation rate (SAR) against null hypothesis significance levels (H_0 : no orbital forcing); and (ii) to evaluate the reliability of our astrochronological interpretation. Such tests were performed by removing different long-term trends with a 'lowess' smoother for three distinctive sliding windows (4, 6 and 8 m). Significance levels were estimated by using Monte Carlo simulation (5000 iterations) and tested sedimentation rates range from 0.05 to 3 cm/kyr with a step of 0.01 cm/kyr using the software Acycle v 2.4.1. The results are provided in Supplementary material "COCO/eCOCO tests" (Figs. 2–4).

The COCO/eCOCO results support our previous cyclostratigraphic interpretations for the PLG core, based on the ~405-kyr tuned age model for the long eccentricity low-pass filter output from MS data. Our sedimentation rate curve (Fig. 4a - main text) give a mean SAR of ~0.44 cm/kyr for the most of Aptian. This result is consistent with the COCO/eCOCO results (two peaks of 0.52 cm/kyr and 0.58 cm/kyr associated to a null hypothesis significance level lower than 0.001 - Supplementary Figs. 2, 3 and 4).

We discuss these results in the main text: "*So far, the only estimate of mean sedimentation rate inferred for the entire Aptian (based on the PLG core) in literature is ~ 0.24 cm/kyr³². In contrast, our sedimentation rate curve (Fig. 4a) indicates a mean SAR of 0.44 cm/kyr for the most of Aptian. This result is consistent with the COCO/eCOCO results (two peaks of*

0.52 cm/kyr and 0.58 cm/kyr associated to a null hypothesis significance level lower than 0.001 - Supplementary Figs. 2–4).” (L. 281–285)

Comment 2:

Moreover, eFFT should be explained when it is first used in the paper.

Response:

We explained about eFFT the first time it is used into the main text: “Comparing these spectral peak frequencies with evolutionary Fast Fourier transform (eFFT) results (Fig. 3, bottom), we observed a pronounced low-wavelength stability pattern throughout the section.” (L. 188–189)

Response to Comments from Reviewer #2**General Comments:**

This manuscript presents multi-proxy records from a core that spans a geochronologically controversial time interval, and uses astronomical tuning to address geochronological uncertainties especially as they pertain to stage boundaries and ancient climatic events. The results generated new age and duration estimates that compliment previously published dates that used radiometric methods.

The methodology, data analysis, interpretation, claims and conclusions appear sound. Although, most major conclusions are tied to geochronology, they should be of interest to a wide range of earth scientists since they concern the geologic timescale.

Response:

Thank you for your comments and recognition of the importance of our work.

Reviewers' Comments:

Reviewer #1:

Remarks to the Author:

The newly added statistical tuning results generally support the mean sedimentation rate is 0.44 cm/kyr and the eCOCO results demonstrate the robustness of the interpretation. Congratulations!

I have no more comments. Looking forward to seeing the publication of this nice work.